# A scoping review of substance use brief interventions in Africa

**Florence Jaguga**[1]*, **Sarah Kanana Kiburi**[2], **Eunice Temet**[3], **Matthew C. Aalsma**[4], **Mary A. Ott**[5], **Rachel W. Maina**[6], **Juddy Wachira**[7], **Cyprian Mostert**[8,9], **Gilliane Kosgei**[10], **Angeline Tenge**[7], **Lukoye Atwoli**[6,9]

**1** Department of Alcohol and Drug Abuse Rehabilitative Services, Moi Teaching & Referral Hospital, Eldoret, Kenya, **2** Department of Medicine, Aga Khan University Hospital, Nairobi, Kenya, **3** Department of Psychiatry, Moi Teaching & Referral Hospital, Eldoret, Kenya, **4** Department of Pediatrics, Division of Child Health Services Research, Adolescent Behavior Health Research Program, Indiana University, School of Medicine, Bloomington, Indiana, **5** Department of Global Health and Health System Design, Arnhold Institute for Global Health, Icahn School of Medicine at Mount Sinai, New York, New York, United States of America, **6** Department of Medicine, Medical College East Africa, The Aga Khan University, Karachi, Pakistan, **7** Department of Mental Health and Behavioral Sciences, School of Medicine, Moi University, Eldoret, Kenya, **8** Department of Population Health, Medical College East Africa, The Aga Khan University, Karachi, Pakistan, **9** Brain and Mind Institute, The Aga Khan University, Karachi, Pakistan, **10** Afya ya Akili Mashinani, Academic Model Providing Access to Healthcare, Eldoret, Kenya

* flokemboi@gmail.com

**Data Availability Statement:** All relevant data are within the manuscript and its Supporting Information files.

**Funding:** The authors received no specific funding for this work.

## Abstract

### Background

The burden of substance use in Africa is substantial. Brief interventions (BIs) are a recommended public health strategy for the prevention and early intervention for substance use problems. The objective of this scoping review was to map the literature on substance use BIs in Africa, identify gaps, and provide directions for future research.

### Methods

The scoping review was guided by the Arksey O'Malley Framework and the PRISMA-Scoping review (PRISMA-ScR) guidelines. A search of five bibliographic databases (PubMed, PsychINFO, Web of Science, Cumulative Index of Nursing and Allied Professionals (CINAHL) and Cochrane Library) was conducted from inception until 1st November 2023. BIs were defined as those targeting substance use and delivered over 1–4 sessions, or interventions delivered over more than four sessions if the authors referred to them as 'brief'. Results of the review have been summarized descriptively and organized by three broad outcomes: BI effect and feasibility; feasibility and effectiveness of training providers to deliver BIs; Other outcomes i.e. cost-effectiveness, BI adaptation and development, and knowledge attitude and practice of BIs by providers.

### Results

Of the 80 studies that were eligible for inclusion, 68 investigated the effect and feasibility of BIs, six studies investigated the feasibility and effectiveness of training providers to deliver BIs, and six explored other outcomes. Most of the available studies had been conducted in

**Competing interests:** The authors have declared that no competing interests exist.

Eastern and Southern Africa. BIs were largely based on motivational interviewing and psychoeducational principles. Overall, the BIs were reported as feasible to implement from the perspective of policy makers, providers, and the intervention recipients. Findings on the effect of BIs on substance use were mixed. Key evidence gaps emerged. There was paucity of BI research focusing on substances other than alcohol, and there was limited literature on feasibility and efficacy of BIs among youth and adolescents.

## Conclusion

The results of this scoping review provide important directions for future substance use BI research in Africa.

## 1. Introduction

Psychoactive substances are those which when taken into the system affect mental processes such as mood, consciousness, and cognition [1] Examples include alcohol, tobacco, stimulants (amphetamines, methamphetamines, khat, cocaine), opioids, cannabis, inhalants, sedative/hypnotics, hallucinogens [1] Even though the literature makes a distinction between 'non-pathological' or 'safe' levels of substance use, and substance use disorders (SUDs) [1,2], the current understanding is that there are no safe levels of substance use [3–5].

Substance use and SUDs are a significant public health problem in Africa. According to the 2023 World Drug Report [6], the past-year prevalence of cannabis use was 10% (30 million people) in 2021, while in the same year, the estimated prevalence for opioid use was 1.2% within Africa [6]. The use of new psychoactive substances is on the rise in Africa, and close to one million people inject drugs (i.e. introduce drugs such as opioids, cocaine, and amphetamines through their veins into the bloodstream) within the continent [6]. In a systematic review and meta-analysis, the authors reported that the pooled prevalence of alcohol use disorders in sub-Saharan Africa was 9.5% [7].

The burden associated with substance use and SUDs in Africa is high. In 2019, the aged-standardized alcohol-attributable deaths and Disability Adjusted Life Years (DALYs) were second highest for the World Health Organization (WHO) African Region at 52.2 deaths and 2182 DALYs per 100 000 people [8]. Alcohol-attributable burden of disease refers to the burden associated with "...health conditions that are entirely attributable to alcohol use (i.e. when alcohol use serves as the necessary and sufficient causal factor for their development, and the sole or main cause of their development), as well as health conditions which are partially attributable to alcohol consumption (i.e. when alcohol use serves as an important and preventable risk factor which increases the probability of development of such conditions or associated mortality and disability, although these health conditions may also occur or develop without any exposure to alcohol)" [8].

Despite such a high burden, opportunities for SUD treatment remain limited in Africa [9,10]. In fact, the treatment gap for SUDs in Africa is as high as 87% [11]. Brief interventions (BIs) are a well-established strategy for the prevention and early intervention for substance use and SUDs and are recommended by important authorities such as the WHO [12] and United Nations Office on Drugs and Crime (UNODC) [13]. Click or tap here to enter text. The goal of BIs is to encourage those with low to moderate risk substance use, to reduce or stop using substances, and those with high-risk use to engage with specialist care [12]. Screening is often done prior to BIs to identify those needing the intervention. BIs are generally based on

motivational interviewing (MI) techniques and their goal is to encourage a change in the pattern of substance use [12].

BIs represent a scalable public health approach to addressing substance use problems in Africa for several reasons. Firstly, BIs are time-limited, and can be delivered in various settings including primary healthcare [14], criminal justice systems [15], and schools [16]. Secondly, BIs can be delivered by a wide range of providers including primary healthcare workers [17], lay providers [18] and through digital means [19]. Most importantly, meta-analytic evidence indicates that BIs are effective for treating harmful substance use [14,20]. It is however important to note that the effectiveness and acceptability of BIs for addressing substance use problems depends on several factors such as delivery setting [21], substances used [22], and number of BI sessions [23].

Over the years, investigators in Africa have worked to explore various outcomes related to substance use BIs [18,24–26]. Little has, however, been done to synthesize and summarize the substance use BI work done in this region. In one review paper, the authors summarized literature on substance use BIs in sub-Saharan Africa. The paper however summarized work done between 1980 and 2009 and focused on alcohol BIs only [27].

The aim of this scoping review is to build upon this prior work by (i) mapping existing literature on substance use BIs in Africa, (ii) describing the characteristics of the published studies on substance use BIs in Africa, (iii) describing the findings of the published studies on substance use BIs, (iv) identifying areas where there is limited research evidence and making specific recommendations for future substance use BI research, and to (v) describe the components of BIs in Africa.

The ultimate goal of this work is to spur further substance use BI research in Africa. This work aligns with the target 3.5 of the Sustainable Development Goals which requires that countries strengthen substance use treatment and prevention [28].

## 2. Methods

This was a scoping review of primary literature on substance use BI-related research in Africa. The scoping approach fit in with the aim of our review which was to explore the breadth of literature on BIs in Africa, as well as describe the definition of BIs [29]. The scoping review was guided by the Preferred Reporting Items for Systematic reviews and Meta-Analyses for Scoping reviews (PRISMA-ScR) guidelines, and the five step Arksey O'Malley Framework [30] which has been updated by Levac and colleagues [31].

The five steps include:

1. Identifying the research question

2. Identifying relevant studies

3. Study selection

4. Charting the data

5. Collating, summarizing, and reporting results

### 2.1 Stage 1: Identifying the research question

Our goal was to document all primary research articles from Africa describing various outcomes of substance use BIs such as acceptability, effectiveness, feasibility, and cost-effectiveness. Following the conception of the topic, an exploratory search was carried out to determine the extent of literature on substance use BIs in Africa. The exploratory search was

conducted in PubMed using terms for substance use, Africa, and BIs. Following this exploratory search, we noticed that the definition of BI was heterogeneous across articles with sessions ranging from 1–7, and that interventions utilized varied strategies including cognitive behavioral therapy [32], problem solving therapy [33]MI [25], and advice [34] either offered alone or in combination. We observed that the terms used to refer to BIs were diverse and included MI, brief treatment, screening and BI, screening BI and referral to treatment (SBIRT), and brief counselling, and these were included in the final search strategy. The exploratory search also helped with identifying the search terms to be used for "substances". We also observed that the study outcomes were varied and included BI effect on substance use; BI acceptability and feasibility from the perspective of various stakeholders; effectiveness and feasibility of training providers to deliver the BI; BI cost-effectiveness. Because this was a scoping review whose goal was to explore the breadth of substance use BI literature in Africa, the authors agreed to include studies investigating all the aforementioned outcomes. Because of the heterogeneity in the definition of BIs across studies, the authors decided that an additional aim of the scoping review would be to provide a description of what authors referred to as substance use BIs within the African literature.

## 2.2 Identifying relevant studies

Five different electronic databases: PubMed, PsychINFO, Web of Science, Cumulative Index of Nursing and Allied Professionals (CINAHL) and Cochrane Library were used to search for articles published in English or translated to English to identify relevant studies. Different search engines were engaged, and the initial database searches were conducted from October 31st to 1st November 2023.

There was no filter applied for the year of publishing, and all relevant studies up to date were assessed. We conducted a hand search of references for identified articles to check for additional papers of BIs but did not find any over above those that we had obtained from the database searches. The keywords used for the search in this review were related to substance use, BIs, and Africa. The same terms were used across all databases. The terms we used for substances were obtained from the exploratory search and from evidence on commonly used substances in Africa [35]. The search terms have been provided in S1–S5 Files.

**2.2.1 Inclusion criteria.** We included articles that explored BIs targeting substance use if: (a) the population examined or part of the population was from Africa, (b) the article was primary research, (c) articles were published in English or had an English translation available.

Because this was a scoping review, we included articles that examined a broad range of outcomes such as acceptability, feasibility, efficacy, effectiveness, cost-effectiveness, BI adaptation, and provider knowledge and practice of BIs. We included studies that utilized all study designs including qualitative, quantitative, and mixed methods. Additionally, scoping reviews allow for decision making and flexibility during reviewing articles for inclusion. For example, for effectiveness trials with a control arm, we agreed to include studies that had the BI as control condition. For this study, BIs were defined as those delivered over 1–4 sessions as described by Mattoo et al [36]. whether the authors referred to them as brief or not, or interventions delivered over more than four sessions if the authors referred to them as 'BIs'. We settled on this definition following a preliminary search on PubMed that showed that there was significant heterogeneity in the way studies had defined substance use BIs. Although some authorities like SAMHSA provide a definition that differentiates brief therapies from BIs, they acknowledge that the distinction in practice can be unclear [37].

**2.2.2 Exclusion criteria.** Studies were excluded if: (a) they were cross-national and did not report specific results for Africa, (b) they were review articles, dissertations, conference

presentations or abstracts, case studies, commentaries, editorials, grey literature, or protocols and, (c) the full text articles were not available (d) the BI did not target substance use.

## 2.3 Stage 3: Literature selection

**2.3.1 Review of abstracts and titles.** Following the search, all articles identified were exported to Mendeley reference manager where the initial removal of duplicates was done. Next, they were exported onto Rayyan[TM] (a software for screening and selecting studies for systematic and scoping reviews and detecting duplicates), [38] where further removal of duplicates was done. Rayyan was additionally used for screening of abstracts and titles of retrieved articles. This stage of screening was independently done by two authors (S.K and F.J) based on the predetermined eligibility criteria. Disagreements during this stage of the screening were resolved through discussion and consensus with a third author (E.T.). For articles that met criteria for inclusion, we retrieved the full texts. All efforts were made to contact authors whose full texts were not available online.

**2.3.2 Full text review.** The second round of screening was done independently by two other authors (G.A. and F.J) and resulted in a 91% agreement. Disagreements during this stage were resolved through discussion and consensus. In instances where consensus could not be reached, a third author was invited to review (E.T.). For this second review, a record was kept for excluded studies and the reasons for exclusion.

## Stage 4: Charting the data

A data extraction form was prepared in Microsoft Excel by F.J. The form was first piloted independently by F.J and E.T using ten included articles. The two authors then met to determine whether their approach to data extraction was consistent with the research question and purpose.

Piloting was done to ensure consistency, and necessary adjustments were made to the form thereafter. Relevant data was extracted and entered into the form by all authors. F.J. double checked the completed form for completeness and accuracy. The following data were extracted: author, year, country, study design, sample size, study setting, study population, age and gender distribution, BI details (screening tool, BI components, mode of delivering the BI, targeted substance, mode of delivery, whether it was described by authors as 'brief' or not), study outcomes and key findings.

The results of the search have been presented using a search decision flow chart and a narrative describing the process. The flow chart will follow the format given by PRISMA-ScR guidelines [39]. A completed PRISMA-ScR checklist has been provided in (S1 Checklist).

## 2.5 Collating, summarizing, and reporting results

After familiarization with the data, two authors (F.J and E.T.) inductively identified three main themes regarding BI outcomes. The themes were reviewed and affirmed by the other authors and are listed below:

1. BI effect and feasibility (n = 68). These were further organized into:

   - Single session BIs (n = 32)

   - Multi-session BIs (n = 30)

   - BIs with the number of sessions not described (n = 6)

2. Feasibility and effectiveness of BI provider training (n = 6)

3. Other BI outcomes e.g. cost-effectiveness analysis, BI adaptation, knowledge attitude and practice on BIs (n = 6).

Findings have been organized as a narrative review based on these themes. A discussion section providing implications of findings has been provided.

## 2.6 Protocol registration and quality assessment

There was no formal registration of this scoping review with PROSPERO™, the international systematic review database. As of the time of writing this manuscript, it is not a requirement for scoping reviews to be registered with PROSPERO™. Further we did not conduct a quality assessment since this is not mandatory for scoping reviews [40].

# 3. Results

## 3.1 Search results

The search from the five electronic databases yielded 1409 results: 659 from PubMed, 138 from PsycINFO, 239 from web of science, 98 from CINAHL and 275 from Cochrane library. A total of 435 duplicates were identified and removed, and 974 studies remained and underwent an initial screening based on abstracts and titles. Of these, 872 articles were excluded either because they did not meet eligibility criteria (n = 860), or full texts were not found (n = 12). A second screen of full text articles was done for the 102 studies that were eligible for the review. Following the second round of screening, 22 studies were excluded because they did not meet eligibility criteria as follows: intervention not meeting our definition of 'brief' n = 15; study not conducted in Africa n = 4; multinational study with no separate data on Africa n = 1; intervention not targeting substance use n = 2. A total of 80 studies were found to meet the inclusion criteria and were included in the review (Fig 1).

## 3.2 General description of studies

**3.2.1 Country.** The studies were conducted across 12 African countries. The highest number of studies had been conducted in South Africa (n = 49), followed by Kenya (n = 9), and Uganda (n = 6). The other studies were conducted as follows: Nigeria (n = 4), Zambia (n = 3), Tanzania (n = 3), and one each for Namibia, Ethiopia, Zimbabwe, Malawi, Mozambique, and Sudan.

**3.2.2 Trends in publication.** The earliest identified study meeting our inclusion criteria was published in 2003 [41]. Thereafter, the number of studies remained low at 1–4 studies per year until 2013 when the number of studies experienced a significant rise. The highest number of studies were published in 2022 (n = 12) (Fig 2).

**3.2.3 Sample size.** The sample sizes ranged from nine, in a study exploring the effectiveness of training lay providers to deliver a BI [42], to 54,187 in a cross-sectional study exploring the feasibility of implementing BI in a health facility and community setting [43].

## 3.3 Effect and feasibility of BIs

**3.3.1 Effect and feasibility for single session BIs.** *Description of studies:* Thirty-two studies explored intervention effects and or feasibility for single session BIs. The study populations included general adult patients (n = 10), people living with HIV (PLWH) (n = 4), patients with tuberculosis (TB) (n = 3), pregnant women and mothers (n = 4), youth (n = 2), patients with mental disorders (n = 1), general community samples (n = 3), male khat chewers (n = 2), patients with hypertension and diabetes (n = 1), providers (n = 2). The study designs included

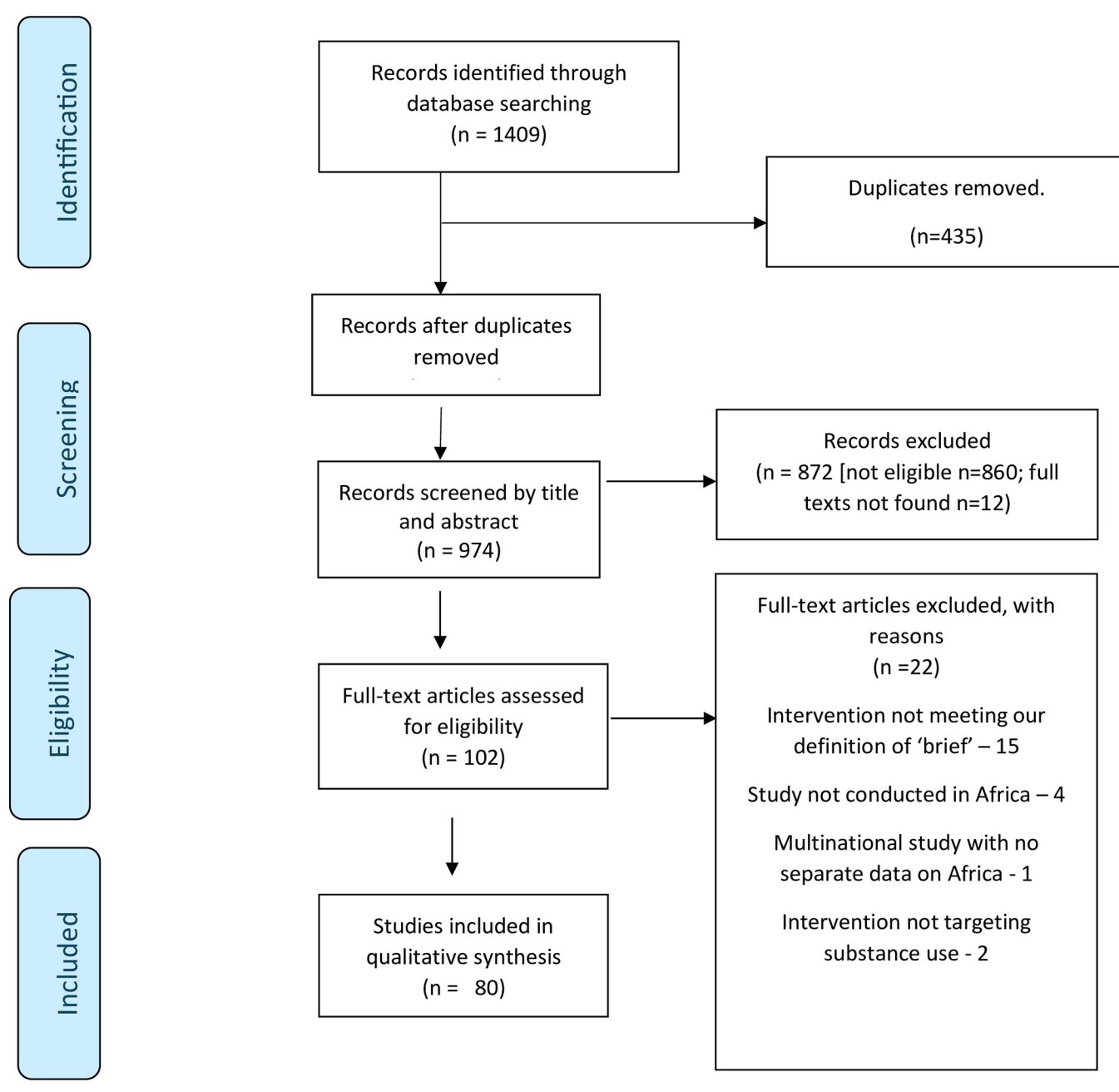

**Fig 1. PRISMA ScR flow chart.**

randomized controlled trial (RCT) (n = 16), quasi-experimental designs (n = 5), mixed methods (n = 5), qualitative (n = 3), cross-sectional (n = 2), and retrospective cohort (n = 1).

Sample sizes for the RCTs ranged from 40 [44] to 1236 [45]; for the qualitative studies from 13 [46] to 52 [47]; for the cross-sectional studies from 18 [48] to 54,187 [43]; for the mixed methods studies from 40 [49], to 1199 [50] and for the quasi experimental studies from 127 [51] to 1203 [52]. One retrospective cohort study had a sample size of 333 [53].

The single session BIs targeted alcohol only (n = 16), multiple substances (n = 9), tobacco only (n = 5), and khat (n = 2). The sessions were delivered by specialist mental health providers

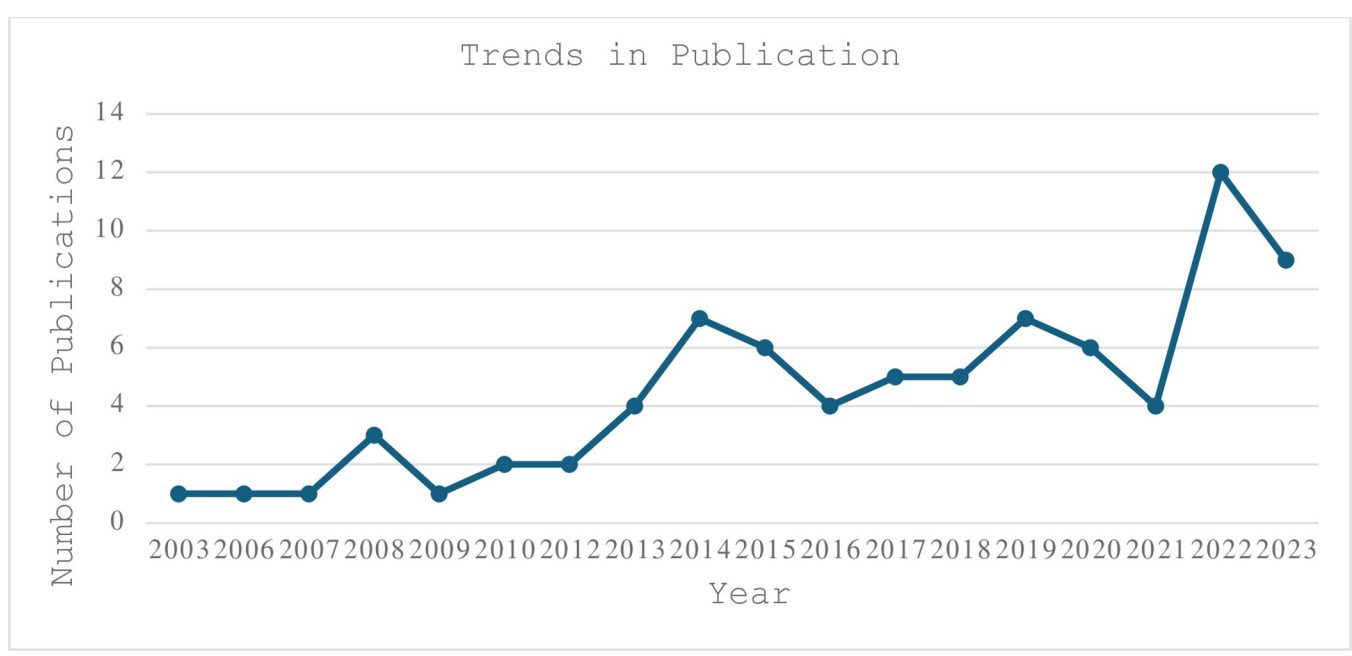

**Fig 2. Trends in publication of BI literature.**

(e.g. psychologists), non-mental health providers (e.g. nurses, physicians, social workers) and lay providers (e.g. persons in recovery). Most sessions were delivered in-person (n = 30). In one study, the intervention was delivered either face to face or by mobile phone [54]. Another intervention was delivered through a self-administered app [19]. Other study details are in Table 1.

**Findings on BI effect**: Single session BIs resulted in reduced khat use at 1 month but with no further effect between month 1 and 2 [57,70]. The BI resulted in reduced tobacco use at month 3 [55,56], and at 6 months [56,62]. Louwagie et al. [61] found no BI effect on tobacco use at 1 month for those with high nicotine dependence (Table 1).

Concerning alcohol use, single session BI resulted in reduced use at month 1 [50,54]; 6 weeks [72], 3 months [50,61–64], and 6 months [50,52,60,72]. Other studies reported no intervention effect on alcohol use at month 3 [44], month 5 [58], month 6 [59,64,68] and month 12 [58,64]. Findings for the effect of BI on multiple substances were mixed. Some studies found no effect on total Alcohol Smoking and Substance Involvement Screening Test (ASSIST) scores [62,68], while others reported a statistically significant reduction in total ASSIST scores [49,69]. Mertens et al. [63] reported that the BI had no effect on cannabis, and methamphetamine use at month 3 (Table 1).

**Findings on BI feasibility**: The intervention recipients reported that the BI provided motivation to quit [46,73]. The BI was reported as appropriate and acceptable by patients, caregivers, and providers [19,73]. Barriers to implementation from providers' perspectives included time and space [47,49], competing work priorities [48], and lack of referral pathways ([71] (Table 1).

**3.3.2 Effect and feasibility for multi-session BIs.** *Description of studies*: Thirty studies explored intervention effects or feasibility for multi-session BIs. The study populations included general adult patients (n = 3), PLWH (n = 12), patients with TB (n = 5), pregnant women and women of childbearing age (n = 2), youth and adolescents (n = 4), general community samples (n = 3), female sex workers (n = 2), patients with chronic disease (n = 2),

**Table 1. Summary of studies investigating effect and feasibility of single session BIs.**

| Author, year, country | Study design | Study population, Setting sample size/gender/age | Description of single session intervention | Control intervention/additional study arm | Quantitative outcomes and findings | Qualitative outcomes and findings |
|---|---|---|---|---|---|---|
| **Quasi-experimental studies** | | | | | | |
| 1. Everett-Murphy 2010 [55] South Africa | Quasi-experimental | **Setting:** Healthcare facility (antenatal care clinics) **Study population:** Pregnant women attending ANC clinic. **Sample size:** 979 self-reporting pregnant smokers (443 in the control; 536 in the intervention group). **Gender:** all female **Mean age:** Control arm 24.0 (SD 6.0); interventions arm 24.1 (SD 6.0) | **Screening:** Single question asking about smoking status **BI:** 5As (Ask, Advice, Assess Assist, Arrange) Guideline for brief smoking cessation **Number of sessions:** 1 **Duration:** Not indicated **Substance:** Cigarette smoking **Delivery:** Peer counselors (persons who had quit smoking); midwives; In–person. **Intervention described as brief:** Yes | Treatment as usual | **Primary outcome:** quitting i.e. urinary cotinine level below 100 ng/ml measurements taken at baseline (i.e. less than 24 weeks gestation, mid-pregnancy (28–35 weeks gestation), and late pregnancy at 36–39 weeks. **Findings:** Difference in quit rates between the two cohorts in late pregnancy was 5.3% (95% CI: 3.2–7.4%, $p < 0.0001$) in an intention to treat analysis. **Secondary outcome:** reduction in smoking defined as at least half the level of urinary cotinine at study entry, and self-reported quitting, reduction, and quit attempts. **Findings:** There was a significant difference in reduction of smoking of 11.8% (95% CI: 5.0–18.4%, $p = 0.0006$). | None |
| 2. Lasebikan 2017 [52] Nigeria | Quasi-experimental | **Setting:** Community **Study population:** adult alcohol users. **Sample size:** 1,203. **Age:** 508 participants were < 25 years **Gender:** Male-623 | **Screening:** ASSIST- **BI:** brief MI **Substance:** Alcohol **Delivery:** Lay providers; In-person **Intervention described as brief:** yes | None | **Primary outcome:** Change in the ASSIST alcohol use scores from baseline to follow-up. **Findings:** There was a statistically significant difference in the prevalence of alcohol use at baseline compared with that at 6 months, $\chi2 (2) = 4.2$, $p = 0.01$. Mean ASSIST scores significantly reduced between time points [$F(1.541, 34.092) = 53.241$, $p < 0.001$]. | NA |
| 3. Lasebikan 2016 [56] Nigeria | Quasi-experimental | **Setting:** Community **Study population:** Male and female tobacco users of age ≥15 years **Sample size:** n = 1203 **Mean age (SD):** 24.45 ± 9.23 years **Gender:** Male 623 | **Screening:** ASSIST **BI:** **Low risk use-** general health advice **Moderate risk use-** MI and leaflet containing information about tobacco use. **High risk use-** MI; leaflet containing information about tobacco use, and referral for treatment **Substance:** Tobacco-cigarette smoking **Mode of delivery:** lay providers; In-person **Duration:** not mentioned **Intervention described as brief:** yes | None | **Outcomes:** Changes in the mean number of cigarettes smoked/day at 3- and 6-months post-intervention and the mean ASSIST scores at 3 and 6 months. **Findings:** At baseline, out of 1203 respondents, lifetime prevalence and current prevalence of any tobacco products were 405 (33.7%) and 248 (20.6%), respectively. At 3 months, out of 1199 respondents, prevalence of current users was 199 (16.5%) and out of 1195 respondents, was 169 (14.1%) at 6 months. Prevalence of tobacco use reduced significantly at 3 months $Z = −3.1$, $p = 0.01$ and at 6 months when compared with baseline $Z = 4.2$, $p = 0.001$, but not at 6 months compared with at 3 months, $Z = 2.1$, $p = 0.09$. | None |
| 4. Sorsdahl, 2012 [51] South Africa | Quasi-experimental | **Setting:** Healthcare facility **Study population:** adult patients seeking psychiatric services. **Sample size:** 127 **Gender:** 88% were male **Age:** Mean age: 30 years. | **Screening:** ASSIST **BI:** MI and referral to specialist care **Duration:** Not indicated **Substances:** Multiple **Delivery:** Social worker; In-person **Intervention described as brief:** Yes. | None | **Primary outcome:** Change in substance use scores. **Findings:** Substance-use involvement scores decreased significantly following intervention (pre-intervention mean 37.60±8.433, post-intervention mean 17.02±17.19, $t(72) = 10.89$, $p<0.001$). Reductions were observed in the use of all classes of drugs except cocaine ($p = 0.742$). **Other outcomes:** Service satisfaction, barriers to treatment, reasons for not accessing further treatment. **Findings:** Of those that attended treatment services, 55% felt that the facility met their needs, 27% that the facility mostly met their needs, and 18% that the facility did not meet their needs. | None |

(*Continued*)

**Table 1.** (Continued)

| Author, year, country | Study design | Study population, Setting sample size; gender/age | Description of single session intervention | Control intervention/additional study arm | Quantitative outcomes and findings | Qualitative outcomes and findings |
|---|---|---|---|---|---|---|
| 5. Widman 2022 [57] Kenya | Quasi-experimental | **Setting:** Community **Study population:** male Somali khat chewers. **Sample size:** n = 330 (n = 161 BI group; n = 169 Control group) **Gender:** all male **Age:** Mean age was 27.8 years (SD 6.6) | **Screening:** ASSIST **BI:** MI **Number of sessions:** 1 **Duration:** 5–10 minutes **Substance:** Khat use **Delivery:** College graduates, in person **Intervention described as brief:** Yes | Screening with ASSIST only | **Outcomes:** Change in khat use **Findings:** Over the 2-month observation period and from baseline to month 1, khat use amount and frequency decreased (p < .001) and the intervention group showed a greater reduction (group x time effects with p ≤ .030). From month 1 to month 2, no further reduction and no group differences emerged. **Feasibility:** the study team recruited 330 participants out of 366 approached khat users, showing that counseling to reduce khat use was acceptable. The dropout rates in BI groups were small supporting the high acceptance of the BI in the target population | None |

**Randomized controlled trials (RCT)**

| Author, year, country | Study design | Study population, Setting sample size; gender/age | Description of single session intervention | Control intervention/additional study arm | Quantitative outcomes and findings | Qualitative outcomes and findings |
|---|---|---|---|---|---|---|
| 6. Harder 2020 [54] Kenya | RCT | **Setting:** Healthcare facility (a primary health center in rural Kenya) **Study population:** adult participants who screened positive for alcohol use problems. **Sample size:** 300 (mobile MI (n = 89), in-person MI (n = 65), or delayed mobile MI (n = 76) for waitlist controls one month after no treatment, with 70 unable to be reached for intervention.) **Gender:** Male ~78% **Age:** Mean age: 38 years | **Screening:** AUDIT **BI:** Two arms: mobile MI, and standard in-person MI **Duration:** 30 mins **Delivery:** Health center staff in the HIV clinics; by mobile phone, or in-person at the health center **Substance:** Alcohol **Intervention described as brief:** Yes. | One-month waitlist followed by delayed mobile MI | **Primary outcome:** Comparison of change in average drinking scores for waitlist vs. mobile MI **Findings:** AUDIT-C scores were nearly three points higher (Difference = 2.88, 95% CI: 2.11, 3.66) for waitlist controls after one month of no intervention vs. mobile MI one month after intervention. **Secondary outcome:** comparison of change in average drinking scores for in-person vs. mobile MI **Findings:** There was no difference between in-person and mobile MI at one month (Bayes Factor = .22); results were inconclusive at six months (Bayes Factor = .41). | None |
| 7. Huis in 't Veld, 2019 [58] South Africa | RCT | **Setting:** Health facility (HIV clinic) **Study population:** PLWH **Sample size:** n = 560 **Age:** Mean age (range) 36 years (31–42) **Gender:** Male 53.9% | **Screening:** AUDIT **BI:** Brief MI; plus health education leaflet on responsible drinking, control group received only the leaflet. **Duration:** Not mentioned **Substance:** Alcohol **Delivery:** nurses; In-person **Intervention described as brief:** yes | Health education leaflet on responsible drinking. | **Primary outcomes:** Change in AUDIT scores between baseline and 5- and 12-month follow-up. **Secondary outcomes:** Changes in HIV outcomes (last measured CD4 cell count, last measured HIV viral load and ART adherence); quality of life outcomes (WHOQoL-HIVBREF tool); depression score; level of experienced stigma and tobacco use. **Findings:** There was a significant decrease in total AUDIT scores between baseline and follow up points 1 (5 months) and 2 (12-month) in both groups. There was no significant decrease between time points 1 and 2. However, between the intervention and control groups there was no difference in reduction of alcohol use to abstinence or low risk alcohol use over time as there was no difference in absolute decrease in AUDIT-score or percentage of change in AUDIT score. The intervention had no influence on the quality-of-life outcomes, depression scores, stigma, tobacco use, viral load and therapy adherence at both time points. | None |

*(Continued)*

**Table 1.** (Continued)

| | Author, year, country | Study design | Study population, Setting sample size; gender/age | Description of single session intervention | Control intervention/additional study arm | Quantitative outcomes and findings | Qualitative outcomes and findings |
|---|---|---|---|---|---|---|---|
| 8. | Kalichman 2008 [59] South Africa | RCT | **Setting:** informal alcohol serving establishments **Study population:** Adult in the alcohol serving establishments **Sample size:** n = 353 **Age:** Median age- 34. **Gender:** Male 33.1% | **Screening:** AUDIT **BI:** Brief MI () plus HIV education **Duration:** 3hr (180 mins) **Substance:** alcohol **Delivery:** Lay providers; In-person **Intervention described as brief:** Yes | HIV and alcohol education | **Primary outcomes:** sexual risk and protective behaviors e.g. drinking in sexual contexts. **Findings:** The intervention demonstrated significantly less unprotected intercourse, alcohol use before sex, numbers of sex partners, partners met at drinking establishments and greater condom use relative to the control group. **Secondary outcomes:** risk-reduction self-efficacy, alcohol outcome expectancies, behavioral intentions, HIV-prevention knowledge, and AIDS-related stigmas. **Findings:** However, intervention effects were moderated by alcohol use; lighter drinkers demonstrated significantly more intervention gains than heavier drinkers in the risk-reduction condition. Intervention effects occurred at 3 months follow-up and dissipated by 6 months. | NA |
| 9. | Kane 2022 [60] Zambia | RCT | **Setting:** Healthcare facility (HIV clinic) **Study population:** adults PLWH. **Sample size:** N = 128 (N = 64 per group) **Gender:** 44% female **Age:** average age was 40 years | **Screening:** AUDIT **BI (control arm):** Education on alcohol harms, exploring ways to change or reduce use, understanding motivations for use, skills training. **Number of sessions:** 1 **Duration:** 20–30 min **Substance:** alcohol **Delivery:** In-person by HIV peer educators **Intervention described as brief:** yes | **Intervention:** BI + CETA (multi-session transdiagnostic cognitive behavioral therapy) (6–12 sessions) | **Primary outcome:** Outcomes were measured at baseline and a 6-month follow-up and included change in substance use scores. **Findings:** Statistically and clinically significant reductions in mean AUDIT score from baseline to 6-month follow-up were observed in both groups, however, participants assigned to BI plus CETA had significantly greater reductions compared to BI alone (– 3.2, 95% CI – 6.2 to – 0.1; Cohen's d: 0.48). **Secondary outcome:** depression and trauma symptoms, and other substance use. **Findings:** Significant CETA treatment effects were observed for depression, trauma, and several other substances | NA |
| 10. | Louwagie 2015 [61] South Africa | RCT | **Setting:** Health facility **Study population:** Newly diagnosed adult TB patients who were smokers. **Sample size:** 409 participants (205-treatment group, 204 –control group) **Gender:** All male **Age:** Mean age (SD) 41.3 10.3 | **Screening:** Single question about cigarette smoking. **BI:** brief MI **Duration:** not mentioned **Substance:** tobacco **Delivery:** lay health care workers; In-person **Intervention described as brief:** Yes | Brief counselling from a nurse. | **Outcomes:** Self-reported abstinence; predictors of sustained 3- and 6-month abstinence and 7-day point prevalence abstinence at 1 month. **Findings:** The intervention was ineffective among smokers with high nicotine-dependence at 1 month but was effective for all smokers over longer periods. Higher baseline self-efficacy predicted the 1-month 7-day point prevalence abstinence, but not sustained abstinence. HIV-positive participants' odds of sustained abstinence were about three times higher than those of their HIV-negative counterparts. | |

(Continued)

**Table 1.** (Continued)

| | Author, year, country | Study design | Study population, Setting sample size gender/age | Description of single session intervention | Control intervention/additional study arm | Quantitative outcomes and findings | Qualitative outcomes and findings |
|---|---|---|---|---|---|---|---|
| 11. | Louwagie 2014 [62] South Africa | RCT | **Setting:** Six primary care tuberculosis clinics in a South African township **Study populations:** current male smokers above 18 years, with TB. **Sample size:** 409 (intervention group, n = 205 or brief smoking cessation advice from a TB nurse (control group, n = 204) **Gender:** all male | **Screening:** Heaviness of Smoking Index (HSI) **BI:** MI **Duration:** 15–20 minutes **Substance:** Tobacco **Delivery:** Nurses; In- person **Intervention described as brief:** Yes. | Short standardized smoking cessation message from the TB nurse and a smoking cessation booklet | **Primary outcome:** Self-reported 6-month abstinence; biochemically verified quit rates. **Findings:** Self-reported 6-month sustained abstinence was more than twice as high in the intervention group as in the control group, biochemically verified quit rates were significantly higher in the intervention arm than in the control arm. **Secondary outcomes:** sustained 3-month abstinence. **Findings:** Sustained 3-month abstinence was twice as high in the intervention group as in the control group in both adjusted and unadjusted analysis, the biochemically verified 3-month abstinence rates were significantly higher for the intervention group in the adjusted analysis, but not in the crude analysis. | None |
| 12. | Madhombiro 2019 [44] Zimbabwe | RCT | **Setting:** Health facility **Study population:** PLWH **Sample size:** N = 40 (20 control, 20 intervention group) **Gender:** 17 female and 23 males **Age:** Mean age was 39.5 years (SD = 9.59) | **Screening:** AUDIT **BI:** WHO mhGAP Interventional guideline* (Assessment of alcohol use, brief advice on harmful alcohol use, and referral for probable dependence). **Duration:** 1 h **Substance:** Alcohol **Delivery:** Nurses; In-person **Intervention described as brief:** yes | MI/CBT** (4 sessions) | **Outcomes:** Change in AUDIT scores between baseline and month 3 **Findings:** There was a statistically significant decrease in AUDIT score over time in both groups (p<0.001), however no statistically significant group difference with a mean difference of 0.80, standard error of 2.07 and p = 0.70. | NA |
| 13. | Mertens 2014 [63] South Africa | RCT | **Setting:** Health facility **Study population:** Young adults aged 18 to 24 visiting the hospital. **Sample population:** 403 (206-intervention group, 197- control group) **Mean age of 21 years **Gender:** Male 44% | **Screening:** ASSIST **BI:** MI **Duration:** 10 minutes **Substance:** Multiple **Delivery:** Nurse; In-person **Intervention described as brief:** yes. | Usual treatment and a handout. | **Outcome:** Change in ASSIST scores between baseline and month 3. **Findings:** larger baseline to follow-up reductions in ASSIST scores for the Intervention than the control arm for alcohol, which was the most prevalent substance used. Reductions in alcohol ASSIST scores were 38% in the Intervention arm versus 21% in the control arm. The changes in total ASSIST, cannabis, and methamphetamine scores were not statistically significant. | None |
| 14. | Pengpid 2013 [64] South Africa | RCT | **Setting:** Health facility **Study population:** Adult out-patients. **Sample size:** n = 392 (Control = 196; Intervention = 196) **Gender:** Control: Male– 71.3% Intervention: Male– 3.5% **Age (Mean, SD):** Control: 35.4 (10.5) Intervention: 36.1 (12.4) | **Screening:** AUDIT **BI:** Feedback on AUDIT, health education leaflet, simple advice plus brief counselling about reducing excessive drinking (Information Motivation- Behavioral Skills (IMB) Model **Duration:** 20 min **Substance:** Alcohol **Mode of delivery:** Nurse counsellor; In-person **Intervention described as brief:** Yes | Health education leaflet on responsible drinking | **Outcome:** Change in total AUDIT scores between baseline and month 6 and 12. **Findings:** The results indicated a significant main effect for time, with participants in both study conditions showing reductions in AUDIT scores and heavy episodic drinking scores over time. Further subgroup analysis tested if there was a significant reduction of harmful drinking across treatment groups using multilevel logistic regression. While a trend to reduce harmful drinking in the brief intervention group seems apparent, statistically there was no significant intervention effect [B = 0.06(−0.39 to 0.50) P = 0.808] | None |

*(Continued)*

 

**Table 1.** (Continued)

| | Author, year, country | Study design | Study population, Setting sample size; gender/age | Description of single session intervention | Control intervention/additional study arm | Quantitative outcomes and findings | Qualitative outcomes and findings |
|---|---|---|---|---|---|---|---|
| 15. | Rotheram-Borus 2019 [45] South Africa | RCT | **Setting:** Community **Study population:** Mothers aged 18 years or older and their children from pregnancy to five years post-birth **Sample size:** n = 1236 **Age:** Mean age 26.4 years **Gender:** All female | **Screening:** AUDIT-C **BI:** Education (life-long consequences of alcohol on babies; evaluation of the typical amount of alcohol being used by the pregnant woman when drinking, compared to desirable quantities). **Duration:** not described **Substance:** Alcohol. **Delivery:** Lay providers; In-person **Intervention described as brief:** Yes. | None | **Outcomes:** Mothers' alcohol use (AUDIT-C and AUDIT scores) Child Measures: Growth (Weights and heights); Cognitive functioning at 18 months; Social behavior at 3 years and 5 years **Findings:** Drinking increased over the 5 years post-birth, but was significantly lower in the intervention condition. Compared to abstinence, mothers' problematic drinking was associated with decreased child weight, increased child aggressive behavior, and decreased child executive functioning at 3 and 5 years. The intervention's effect was associated with increased child aggression but decreased the effect of problem drinking on children's aggressive acts and executive functioning. | NA |
| 16. | Sheikh 2016 [65] Zambia | RCT | **Setting:** Health facility **Study population:** Adults. **Sample size:** 114 (56 non-intervention group and 58 in the intervention group) **Gender:** Male 96.5% **Age: Mean age (range)** 32 years (18–65 years) | **Screening:** AUDIT **BI:** WHO mhGAP intervention (Assessment of alcohol use, brief advice on harmful alcohol use, and referral for probable dependence). **Duration:** 20 min **Substance:** Alcohol **Delivery:** Psychosocial counsellor who had a diploma in counselling; In person **Intervention described as brief:** yes | Control group-given treatment as usual (detoxification with diazepam and vitamin supplementation. | **Outcomes:** Number of days of abstinence following discharge. **Findings:** The intervention group had an average time to first drink of 51 days (SD = 14), while the non-intervention group had average time to first drink of 10 days (SD = 16). There was a significant difference between the two groups with the intervention group having a long abstinence period (t = 14.368; df = 112; p = 0.001). | NA |
| 17. | Sorsdahl 2017 [66] South Africa | RCT | **Setting:** Healthcare facility (Emergency Department) **Study population:** adults. **Sample size:** n = 335 (MI-113, MI-PST- 112; Control- 110) **Gender:** Male 218 (65.5%) **Age:** Mean age (age-range) - 28 (18–75) | **Screening:** ASSIST **BI:** **MI arm (control)** MI plus a brochure providing information on the effects of substance. **Number of sessions:** 1 for MI only arm **Duration:** 20 min **Substance:** multiple **Delivery:** Counsellors with bachelors-level education; In-person **Intervention described as brief:** Yes. | **Control:** Brochure providing information on the effects of substance use **MI-PST arm (intervention)** Five session MI-PST intervention; a brochure providing information on the effects of substance. | **Outcome:** associations between intervention (MI or MI PST) uptake and patients' demographic characteristics, presenting health condition and level of substance use involvement **Findings:** In adjusted models, being between the ages of 25 and 39 years increased the likelihood (adjusted odds ratio (AOR) 1.91, 95% confidence interval (CI) 1.02–3.57) of accepting an offer of help compared with substance users in the age group 18–24 years. Polysubstance users (AOR 0.28, 95% CI 0.16–0.48) were less likely to accept an offer of help than patients with problematic alcohol use only, while patients with higher ASSIST scores were more likely to accept an offer of help than those with lower scores (AOR 1.04, 95% CI 1.00–1.08). | None |

*(Continued)*

**Table 1.** (Continued)

| Author, year, country | Study design | Study population, Setting sample size; gender/age | Description of single session intervention | Control intervention/additional study arm | Quantitative outcomes and findings | Qualitative outcomes and findings |
|---|---|---|---|---|---|---|
| 18. Sorsdahl 2015 [67] South Africa | RCT | **Setting:** Healthcare facility (Emergency Department) **Study population:** adults. **Sample size:** 335 (113 in the MI, 112 to the MI-PST condition and 110 to the Control arm) **Gender:** Male 218 (65.5%) **Age:** Mean age (age-range) - 28 (18–75) | **Screening:** ASSIST **BI:** MI arm (control) MI; a brochure providing information on the effects of substance use **Number of sessions:** 1 for MI only arm **Duration:** 20 min **Substance:** multiple **Delivery:** Counsellors with bachelors-level education in person **Intervention described as brief:** Yes. | **Control:** Brochure providing information on the effects of substance use **MI-PST arm (intervention)** Five session MI-PST intervention; a brochure providing information on the effects of substance. | **Primary outcome:** Change in ASSIST scores from baseline to month 3. **Findings:** ASSIST scores at three months were lower in the MI-PST group than they were in the MI and control groups (adjusted mean difference of −1.72, 95% CI −3.36 - −0.08). We recorded no significant difference in ASSIST scores between the control and MI group (adjusted mean difference of −0.02, 95% CI −2.01–1.96). **Secondary outcome:** Change in depression scores (CES-D) and frequency of substance-related injury, and violence between baseline and month 3. **Findings:** Participants in the MI-PST arm reported significantly lower CES-D scores relative to combined MI and Control arms at follow-up (t (170) = 2.72, p = p > 0.001) with an adjusted mean difference of 3.33. There were no significant differences in frequency of substance-related injury, and violence between the MI-PST arm and the combined MI and control arms | None |
| 19. Wandera, 2017 [68] Uganda | RCT | **Setting:** Health facility (infectious disease clinic) **Study population:** Adults with HIV **Sample size:** n = 337. **Age:** Median age (IQR)-39(32–46) **Gender:** Males 65.6% | **Screening:** AUDIT-C **BI:** Brief MI **Duration:** 20–30 minutes **Substance:** Alcohol **Delivery:** Trained counsellors (minimum bachelor's degree); In-person **Intervention described as brief:** Yes | Positive prevention counseling (how to prevent infections, nutrition advice, adherence support, encouraging HIV disclosure to partners, safe sex practice and advice on how to avoid alcohol and substance use). | **Outcomes:** Change in alcohol consumption (AUDIT, Timeline follow back) between baseline and month 6. **Findings:** The mean (SD) AUDIT-C scores were 6.3(2.3) and 6.8(2.3), for control and MI arms (p = 0.1) at baseline and change in mean AUDIT-C score was not statistically different between the treatment arms over the 6 months follow up time (P = 0.8). | None |
| 20. Ward 2015 [69] South Africa | RCT | **Setting:** Health facility (Health Centre) **Study population:** Patients aged 18–24 presenting for care. **Sample size:** n = 403 (206 brief MI plus a referral; 197 usual care plus a referral) **Gender and age:** data on gender and age not provided | **Screening:** ASSIST **BI:** MI **Duration:** Not provided **Substance:** Multiple **Delivery:** nurse practitioners; In-person **Intervention described as brief:** yes. | Usual care plus referral resource list | **Outcomes:** ASSIST, Explicit Aggression Scale, and HIV risk behavior scores. **Findings:** The intervention did reduce substance misuse, and reduction in substance misuse was related to reduction in aggression—but reduction in aggression was not greater in the intervention group. | None |
| 21. Widman 2017 [70] Kenya | RCT | **Setting:** Community **Study population:** Male Somalis Khat chewers **Sample size: n =** 330 (n = 161 BI group; n = 169 Control group) **Gender:** all male **Age:** Mean age was 27.8 years (SD 6.6) | **Screening:** (ASSIST) **BI:** Motivational interviewing (MI) **Number of sessions:** 1 **Duration:** 5–10 minutes **Substance:** Khat **Delivery:** College graduates, in-person **Intervention described as brief:** Yes | Screening with ASSIST only | **Outcomes:** Change in mental health symptom severity (Khat use, depression, PTSD) **Findings:** Over the 2-month observation period and from baseline to month 1, khat use amount and frequency decreased (p < .001) and the intervention group showed a greater reduction (group x time effects with p≤.030). From month 1 to month 2, no further reduction and no group differences emerged. Depression and khat-psychotic symptoms, but not PTSD symptoms decreased without group differences. Depression and PTSD did not influence therapy success but in participants with-out comorbid psychopathology, more khat use reduction after the intervention was found (p = 0.024). | None |

**Mixed methods studies**

*(Continued)*

**Table 1.** (Continued)

| Author, year, country | Study design | Study population, Setting sample size/gender/age | Description of single session intervention | Control intervention/additional study arm | Quantitative outcomes and findings | Qualitative outcomes and findings |
|---|---|---|---|---|---|---|
| 22. Clair 2022 [50] Kenya | Mixed methods (RCT and qualitative study) | **Setting:** Health facility (public and private hospitals) **Study population for RCT:** Adult patients seeking care **Sample size:** 1199. **Public hospital: Age:** Mean age 38 years **Gender:** Male- 91%. **Private hospital: Age:** Mean age: 36 years **Gender:** Male- 97% Study population for FGDs and KIIs: 86 patients and 31 health workers, total 117. | **Screening:** ASSIST **BI:** MI **Duration:** Not specified **Substance:** Alcohol **Delivery:** Health care providers; In-person **Intervention described as brief:** Yes. | Feedback only | **Outcomes:** Reduction in alcohol consumption at all points during follow up (1,3, and 6 months); Acceptability/ satisfaction- qualitative data based on FGD s and Key Informant interviews. **Findings:** There was statistically significant reduction in alcohol consumption at almost all time points in both public and private facilities | **Outcomes:** Participant perceptions on the intervention **Findings:** RCT patient participants in both groups reported reduced alcohol consumption with many improvements in mental and physical health and in other dimensions of their lives. Health workers reported that there was a change in their attitudes and practice concerning persons with risky alcohol use. |
| 23. Myers 2012 [49] South Africa | Mixed methods | **Setting:** Health facility (Emergency departments) **Study population:** Peer counsellors and adult patients. **Sample size:** Patients n = 30; emergency room personnel n = 10; sample size not indicated for peer counselors **Age and Gender: Patients:** Female- 18; Mean age (SD)- 29 years (SD = 13) **Emergency room personnel:** Female- 8; age range 26–53 years) **Peer counselors:** age and gender not provided. | **Screening:** ASSIST **BI:** Brief MI **Duration:** 30 minutes **Substance:** Multiple substances **Delivery:** Peer counsellors with bachelor's level of education; In-person **Intervention described as brief:** yes | None | **Outcomes:** patient throughput to assess the feasibility of screening and recruiting patients for an emergency room intervention; patients' **Findings:** During the study period, 1458 patients presenting for emergency services were screened for possible AUD. Of these, 20.9% (n = 305) had moderate to high-risk AUD thus meeting criteria for enrollment. Of these 305 patients, 225 (73.8%) participants were willing to participate. Among the 83 eligible patients who refused the intervention, the two most common reasons for refusing were that they were experiencing too much pain and that they felt they did not have an alcohol problem. | **Outcomes:** peer counselors' perceptions of barriers and facilitators to delivery and scale up of intervention in emergency room settings. **Findings: Barriers:**—lack of private space to conduct the counseling; poor collaboration with emergency room personnel; patient reluctance to disclose alcohol use; nature of patients i.e. some were severely ill. The peers reported the training as adequate. |
| 24. Sorsdahl, 2015 [711] South Africa | Mixed methods (quasi-experimental and qualitative interviews) | **Setting:** Healthcare facility (Obstetric care) **Study population 1:** pregnant women **Sample size included in effect analyses:** 302 who met criteria for depression, 388 who disclosed smoking tobacco, and 29 who disclosed alcohol and other drug use. **Gender:** All female **Age:** Mode- 26 (range 16–46) years **Study population 2:** Healthcare workers and lay providers **Sample size:** 5. **Gender:** not provided **Age:** not provided | **Screening:** Single question asking about smoking status **BI:** 5A's brief intervention. **Number of sessions:** 1 **Duration:** 10–15 min **Substance:** Multiple **Delivery:** Nurses, HIV Counsellors, and a mental health champion; In-person. **Intervention described as brief:** Yes. | None | **Outcome:** Feasibility of Integrating SBIRT into Midwife obstetric services. **Findings:** Over a period of 6 months, 3407 women presented for their first visit. Of these, 1468 (43%) women were screened for maternal mental disorders. Of these, 302 (21.4%) met criteria for depression (based on EPDS), 388 (26.4%) disclosed smoking tobacco, and 29 (2%) disclosed alcohol and other drug use. **Outcome:** Change in EPDS scores between baseline and month 3; **Findings:** EPDS scores decreased significantly following the intervention (pre-intervention mean 18.160 ± 2.5, post-intervention mean 11.94 ± 5.78, t (69) = 8.51, p $\rangle$0.001). **Outcome:** Change in alcohol, tobacco and drug use; **Findings:** Participants significantly decreased their tobacco use (pre-intervention mean 18.160 ± 2.5, post-intervention mean 4.24 ± 1.75, t (73) = 3.45, p$\rangle$0.001). There was no significant reduction in alcohol and drug use | **Outcome:** Providers' responses to the intervention. **Findings:** BI useful for supporting pregnant women's mental health. **Outcome:** Perceptions of barriers and facilitators to the effective delivery of the intervention. **Findings:** Barriers included lack of referral pathways for high-risk women, and women's non-disclosure of alcohol and drug use. |

(Continued)

**Table 1.** (Continued)

| | Author, year, country | Study design | Study population, Setting sample size; gender/age | Description of single session intervention | Control intervention/additional study arm | Quantitative outcomes and findings | Qualitative outcomes and findings |
|---|---|---|---|---|---|---|---|
| 25. | Staton 2023 [72] Tanzania | Mixed methods (RCT plus qualitative interviews) | **Setting:** Health facility (Emergency department) **Study population:** Adults, (>18yrs) **Sample size:** n = 75 **Gender:** 96% male **Age:** Mean (SD) 36 years (13.5) | **Screening:** AUDIT **BI:** Brief MI only **Duration:** 5–30 minutes **Delivery:** Nurses; In-person **Substance:** Alcohol **Intervention described as brief:** yes | 3 comparison groups: Usual care, brief MI with standard text booster, brief MI with personalized text booster. | **Outcomes: Reach** (patient eligibility, recruitment, retention rates; **Effectiveness** (primary- number of binge drinking days; secondary- Drinker Inventory of Consequences (DrInC) score, quantity and frequency of alcohol use, AUDIT score; **Adoption** (patient and provider acceptability); **Implementation** (intervention fidelity; feasibility of randomization processes), **Maintenance** (acceptability of providers to conduct the intervention, perceived time burden, resources required for the intervention). **Findings:** **Reach:** Seventy-five patients were enrolled over the course of 12 weeks; final 6-month retention rate of 84% across all arms. **Effectiveness:** there was a decrease in the DrInC and AUDIT scores across all groups from baseline to 6 weeks, 3 months, and 6 months. At 6-month, participants in the intervention groups showed a higher decrease in average DrInC and AUDIT scores than the usual care group. **Adoption-** The interventions were acceptable with 100% of respondents believing that the BI had a positive impact on their drinking behavior. **Implementation:** Nurses found the BI simple and easy to administer; fidelity rates were high. **Maintenance:** screening for eligibility and providing the BI lasted an estimated 30 minutes per patient. There was agreement between providers and nurses that this time commitment would be feasible for trained nurses to continue in the future. | **Provider acceptability-** Nurses stated they were eager to administer the BI. Nurses believed that most patients enthusiastically participated in and liked the BI; Nurses reported that they felt confident implementing BI in the future. |
| 26. | Tang 2019 [19] Namibia | Mixed methods | **Setting:** Health facility (ART Clinics) **Study population:** PLWH **Sample size:** n = 787 **Age: Mean age (SD)-** 41.2 years (9.6) **Gender:** female 58% | **Screening:** AUDIT **BI:** Brief MI **Duration:** 10–15 minutes **Substance:** Alcohol **Delivery:** Digital (self-administered) **Intervention described as brief:** yes | None | **Outcomes:** Quantitative questions on the usability and acceptability of the intervention. **Findings:** The mean scores for quantitative questions all corresponded to "strongly agree" except one. The participants strongly agreed that the computer program was easy to use, the program was useful, and "I'd rather use the program than talk to a healthcare worker". | **Outcomes:** Qualitative perceptions of patients and providers on the intervention. **Findings:** Overall, the participants found the program to be easy to use and helpful; the healthcare workers reported that the intervention was helpful. |
| **Qualitative studies** | | | | | | | |
| 27. | Lutala 2022 [47] Malawi | Qualitative | **Setting:** Health facilities (NCD clinics) **Study population:** Patients with diabetes or hypertension receiving treatment at the NCD clinic. Nurses, clinical officers, and medical doctors involved in patient management at the clinics. Policy-makers- district health management team **Sample size:** Patients: n = 43, Healthcare providers: n = 6 Policy makers: n = 3 **Mean age:** 36 years old; Range- 26–69 years) **Gender:** not provided | **Screening:** not described **BI:** Behavior change counselling (5As and MI)*** **Duration:** not described **Substance:** alcohol, tobacco **Delivery: Intervention described as brief:** yes. | None | | **Outcome:** Participant's perceptions on the intervention **Findings:** Participants predicted a few challenges such as time and space to conduct the intervention, cultural bottlenecks, low provider- to- patient ratios and high provider turnover. Participants proposed that the intervention is simplified to improve uptake. Participants felt that the intervention could only be sustained if training opportunities are provided and if positive testimonies are given by beneficiaries. Participants perceived that the intervention would contribute to developing the listening ability of healthcare providers, and that it would be affordable, and useful. |

*(Continued)*

**Table 1.** (Continued)

| | Author, year, country | Study design | Study population, Setting sample size/gender/age | Description of single session intervention | Control intervention/additional study arm | Quantitative outcomes and findings | Qualitative outcomes and findings |
|---|---|---|---|---|---|---|---|
| 28. | Petersen 2010 [46] South Africa | Qualitative | **Setting:** Health care facilities **Study population:** Pregnant women. **Sample size:** n = 13 **Age:** Range: 16–40 years | **Screening:** not described **BI:** 5 As for brief smoking cessation counseling **Duration:** not described **Substance:** cigarettes **Delivery:** midwives, peer counselors; In-person **Intervention described as brief:** yes. | None. | | **Outcome:** Qualitative perceptions on the intervention **Findings:** smoking was the norm in the community; the intervention made the women have the desire to quit. Also, the women's confidence to quit increased. |
| 29. | Zewdu 2022 [73] Ethiopia | Qualitative | **Setting:** Healthcare facility (Healthcare center) **Study population:** Adults attending a primary care center with probable Alcohol Use Disorder (AUD); caregivers; non-clinical healthcare workers. **Patients:** Sample size: n = 14 Gender: Male 13 Age: 26–35 years- 5 **Caregivers:** Sample size: n = 4 Gender: Female 4 Age: 26–35 years- 3 **Health workers:** Sample size: n = 8 | **Screening:** AUDIT **BI:** WHO mhGAP Interventional guideline* (Assessment of alcohol use, brief advice on harmful alcohol use, and referral for probable dependence). **Substance:** Alcohol **Duration:** Not specified **Delivery:** Nurses and clinical officers; In-person **Intervention described as brief:** yes | None | NA | **Outcomes:** Participants' perceptions on the intervention **Findings:** Patients and caregivers perceived the intervention to be useful, and acceptable. Patients reported reductions in alcohol consumption and improvements in socio-occupational functioning. |
| **Other designs** | | | | | | | |
| 30. | Goldschmidt 2023 [43] South Africa | Cross-sectional, survey | **Setting:** Healthcare facility, and community setting **Study population:** adults (18 years or older) **Sample size:** n = 54,187 **Gender:** Male 43% **Age:** Median age 34; range (18–85 years) | **Screening:** AUDIT-C **BI:** Brief advice on harms of alcohol use plus referral for high-risk users **Duration:** 10–30min **Substance:** alcohol **Delivery:** lay counselors, In-person **Intervention described as brief:** yes. | None | **Outcomes:** Intervention acceptance among participants **Findings: Acceptance of referral:** Only 9.4% of participants who recorded high scores accepted referrals to local public health clinics. Participant gender, age, and AUDIT-C scores affected rejection of referrals. The odds of referral rejection were higher for older participants and males (OR 1.18; 95% CI 1.16–1.20). The odds also increased with AUDIT-C scores. **Acceptance of brief advice:** Place of screening, gender, age, and AUDIT-C scores all influenced the rejection of brief advice. Men were more likely to reject brief advice than women (OR 1.01; 95% CI 1.01–1.02). Participants with high AUDIT-C scores were more likely to reject brief advice than those with moderate scores (OR 1.16; 95% CI 1.16–1.17). Finally, the odds of rejecting brief advice increased with age. | NA |
| 31. | Peltzer 2008 [48] South Africa | Cross-sectional | **Setting:** Health care facilities **Study population:** primary healthcare facilities. **Sample size:** n = 18 facilities | **Screening:** AUDIT **BI:** Brief MI **Duration:** not described **Substance:** alcohol **Mode of delivery:** nurses; In-person **Intervention described as brief:** yes | | **Outcome:** feasibility and implementation of SBI **Findings:** Over the 6-month period, nurses screened 2670 patients and found that 648 (23.4%) patients (39.1% men and 13.8% women) were hazardous or harmful drinkers. Nine clinics had good and 9 poor SBI implementation. Factors discriminating the clinics with good or poor SBI implementation included the percentage of nurses trained in SBI, support visits, clinical workload, competing priorities, teamwork, innovation adoption curve, perceived complexity of innovation, compatibility beliefs, trialability, and observability of SBI. | NA |

(Continued)

**Table 1.** (Continued)

| Author, year, country | Study design | Study population, Setting sample size; gender/age | Description of single session intervention | Control intervention/additional study arm | Quantitative outcomes and findings | Qualitative outcomes and findings |
|---|---|---|---|---|---|---|
| 32. Reuter 2022 [53] South Africa | Retrospective cohort study | **Setting:** Health care facilities. **Study population:** People living with rifampicin resistant TB. **Sample size:** n = 333 **Median age (range):** 34 years (28–42) **Gender:** Male 58.6% | **Screening:** ASSIST **BI:** MI **Duration:** not described **Substance:** Multiple **Mode of delivery:** lay TB counselors; TB doctors and nurses; In-person **Intervention described as brief:** yes. | | **Outcomes:** Number of patients screened using ASSIST and their TB treatment outcomes (treatment success, loss to follow-up, death) **Findings:** Overall, 333 persons were initiated on TB treatment; 38% (n = 128) were screened for substance use. Of those, 88% (n = 113/128) reported substance use; 65% (n = 83/128) had moderate or high-risk substance use. Seventy-seven persons were screened for substance use within ≤2 months of TB treatment initiation, of whom 69%, 12%, and 12% had outcomes of treatment success, loss to follow-up and death, respectively. | NA |

**Abbreviations:** AIDS- Acquired Immunodeficiency syndrome; ANC- Antenatal Care; ART- Antiretroviral Therapy; ASSIST- Alcohol, Smoking and Substance Involvement Screening Test; AUD- Alcohol Use Disorder; AUDIT-Alcohol Use Disorders Identification Test; AUDIT-C- Alcohol Use Disorders Identification Test-Concise; BI- Brief Intervention; CBT- Cognitive Behavioral Therapy; CES-D- Center for Epidemiologic Studies Depression Scale; CETA- Common Element Treatment Approach; DrinC- Drinker Inventory of Consequences; EPDS- Edinburgh Postnatal Depression Scale; FGDs- Focus Group Discussions; HIV- Human Immunodeficiency Virus; KIIs- Key Informant Interviews; MI- Motivational Interviewing; NCD-Non-Communicable Diseases; PLWH- People Living With Human Immunodeficiency Virus; PST- Problem Solving Therapy; PTSD- Post-Traumatic Stress Disorder; RCT- Randomized Control Trial; SBI- Screening and Brief Intervention; SBIRT- Screening, Brief Intervention, and Referral to Treatment; SD- Standard Deviation; TB- Tuberculosis; WHO mhGAP- Worl Health Organization mental health Gap Action Program; WHOQoL-HIVBREF- World Health Organization Quality of Life- Human Immunodeficiency Virus.

healthcare workers and policy makers (n = 6). Study designs included RCTs (n = 16), quasi-experimental (n = 3), mixed methods (n = 7) and qualitative (n = 4).

Sample sizes for the RCTs ranged from 33 [74] to 1340 [75]; for the quasi experimental studies from 39 [76] to 973 [77]; for the mixed methods studies from 40 [78] to 4,865 [79] and for the qualitative studies from 11 [80] to 49 [81].

Of the studies in which the target substance was indicated, the multi-session BIs targeted alcohol only (n = 19), multiple substances (n = 11), tobacco only (n = 2), and methamphetamine only (n = 1) (Table 2).

The sessions were delivered by specialist mental health providers (e.g. clinical psychologists), non-mental health providers (e.g. nurses, social workers, and medical assistants) and lay providers (e.g. peer educators) (Table 2). Most interventions were delivered in-person (n = 32). Five interventions were delivered using a combination of in-person sessions and digital means [84,85,90,93,99]. One intervention was delivered using a provider-facing mobile health app[100] (Table 2).

*Findings on BI effect*: The interventions resulted in reduced alcohol use at six [85,86,88,96], nine [85], and 12 months [86]. The multi-session BIs also resulted in reduced methamphetamine use at 6 weeks and 3 months [92]. Other studies did not find any BI effect for alcohol use at three [84,89]), six [84], and 12 months [75]. Goedele et al [84] found no BI effect for tobacco use at 3 and 6 months (Table 2).

*Findings on BI feasibility*: The multi-session BIs were reported as useful by the recipients [76,80,83,98,99,101,102], and providers [78,100]. Certain challenges were noted during implementation including competing work priorities, perceived intervention complexity, and lack of teamwork [48]. Goedele et al. [99] reported that the providers felt unprepared for the diversity of problems, attitudes, and beliefs that the patients presented with. O'Grady et al. [100] reported that the provider-facing app ensured an efficient BI session (Table 2).

**3.3.3 Feasibility of BIs with session numbers not described.** Five studies explored the feasibility of BIs but did not specify the number of sessions or session duration (Table 3 rid="_Ref179912376"). Sample sizes ranged from 20 [104] to 1294 [105]. The authors found that BIs were feasible and acceptable from the perspective of student peer mentors [106], PLWH [104], and healthcare workers [107].

Barriers to BI implementation in an emergency department setting included: aggressive patients, lack of private space within the emergency department, severely injured patients [107]. In the study by Miller et al [104], the patients (PLWH) expressed skepticism that participation in a BI intervention would effectively prompt behavior change (Table 3).

## 3.4 Feasibility and effectiveness of BI provider training

*Description of studies*: Six studies explored as their primary aim the feasibility and effectiveness of training providers to deliver a BI. The training participants included healthcare providers [41,48,109–111] and lay providers [42]. Five trainings were conducted face to face [41,42,48,109,110], and one entailed weekly sending of text messages over 13 weeks to physicians on how to conduct the BI [111]. The duration of the trainings lasted from six hours [48,109] to two and a half days [42] (Table 4). Sample sizes ranged from 9 [42] to 946 [111] (for all studies and not by study design).

*Findings*: Three studies quantitatively compared baseline knowledge and skills with those post-training. All studies reported improvements in knowledge and adoption of BIs [109–112]. Three studies explored qualitative perceptions on the intervention. In all studies, the participants reported that the training was useful and improved their capacity to offer SBI [41,42,110]. Several barriers were identified during implementation including challenges in

**Table 2. Summary of studies exploring effect and feasibility for multi-session BIs.**

| | Author, year, country | Study design | Study population, Setting sample size; gender/age | Description of intervention | Control intervention | Quantitative outcomes and findings | Qualitative outcomes and findings |
|---|---|---|---|---|---|---|---|
| **RCTs** | | | | | | | |
| 1. | Bantjes 2023 [82] South Africa | RCT | **Setting:** Health facility (HIV clinics) **Study population:** PLWH **Sample size:** n = 622 **Gender:** Female = 57.5% **Age:** Mean (SD) 41 years (9) | **Screening:** AUDIT **BI:** MI and PST **Number of sessions:** Two **Total duration for all sessions:** Not described **Substance:** Alcohol **Delivery:** HIV clinic staff; In person **Intervention described as brief:** Yes | No Intervention for alcohol use | **Primary outcome:** change in mean AUDIT scores between baseline and at 3- and 6-months follow-up. **Findings:** No significant differences in mean symptoms scores for depression or psychological distress were observed between the intervention and control groups at baseline or at follow-up (p>0.05) **Secondary outcome:** change in mean depression and psychological distress scores between baseline and at 3- and 6-months follow-up. **Findings:** For both the intervention and control groups, there were significant reductions in symptom severity at 3-months and 6-months for depression $F(1,443) = 4.05$, p = 0.0448) and psychological distress $F(1,448) = 9.89$, p = 0.002). Reductions in alcohol consumption were significantly associated with reductions in depression and psychological distress. | N/A |
| 2. | Carney 2019 [83] South Africa | RCT | **Setting:** Community, **Study population:** Female youth between the ages of 16 and 21 who use drugs, are school dropouts and engage in risky sex. **Sample size:** Intervention- n = 206; Control- n = 101 **Gender:** Female only **Age:** Mean age (SD)- 19 years (1.55) | **Screening:** Urine drug testing **BI:** Women health coop (WHC) intervention. based on knowledge and skills to reduce drug use, sex risk and violence with role play and rehearsal **Number of sessions:** 4 modules delivered in 2 sessions in groups of 4–6 women. **Total duration for all sessions:** 4 hours (240 min) **Substance:** multiple substances **Delivery:** trained female interventionists; In-person **Intervention described as brief:** yes | Information about HIV /AIDS, STIs, pregnancy, and sexual risk behaviors. | **Primary outcome:** Changes in urine drug testing and self-reported heavy drinking Changes in sexual behavior. **Findings:** Among participants in the intervention group, there was a statistically significant increase in the proportion of participants testing positive for methaqualone at 1-month follow-up (p < 0.01). For participants in the control group (p = 0.04) and the intervention group (p = 0.07), statistically significant reductions in marijuana were detected. Also, for participants in the intervention group, there was a statistically significant reduction in impaired sex with any partner (p = 0.02) and with main partner (p = 0.06). For the logistic regression analyses controlling for intervention assignment and education, participants in the intervention group were more likely to test positive for methaqualone (adjusted odds ratio [AOR]: 2.32; p = 0.07) and less likely to report having engaged in impaired sex with a main partner (AOR: 0.40; p = 0.07) at follow-up than those in the comparison group. | **Outcome:** Acceptability of the intervention **Findings:** Most FGD participants reported that the intervention was helpful. They reported that they had increased knowledge about alcohol use disorders, STIs, and HIV. |
| 3. | Goedele 2022 [84] South Africa | RCT | **Setting:** Health facility (primary care clinics) **Study population:** Patients with TB **Sample size:** n = 574 (291 control, 283 intervention) **Gender:** Female 22.5% **Age: Mean age (SD)** Intervention 38.56 (11.15) Control 39.37 (12.60) | **Screening:** AUDIT and a single question about tobacco use **BI:** brief MI plus SMS messages (educational plus behavioral skills for medication adherence and alcohol and tobacco use) **Number of sessions:** 3 **Duration:** **Substance:** alcohol, tobacco **Delivery:** In person by lay health workers and digital **Intervention described as brief:** Yes | Treatment as usual | **Outcomes:** Smoking cessation, reduction in AUDIT score; improved TB and ART adherence **Findings:** There was no evidence of an effect at 3 and 6 months, respectively, on continuous smoking abstinence (OR 0.65, 95% CI 0.37 to 1.14; OR 0.76, 95% CI 0.35 to 1.63), TB medication adherence (OR 1.22, 95% CI 0.52 to 2.87; OR 0.89, 95% CI 0.26 to 3.07), taking ART (OR 0.79, 95% CI 0.38 to 1.65; OR 2.05, 95% CI 0.80 to 5.27) or AUDIT scores (mean score difference 0.55, 95% CI −1.01 to 2.11; −0.04, 95% CI −2.0 to 1.91) and adjusting for baseline values. Cure rates were not significantly higher (OR 1.16, 95% CI 0.83 to 1.63). | None |

(*Continued*)

**Table 2.** (Continued)

| | Author, year, country | Study design | Study population, Setting sample size; gender/age | Description of intervention | Control intervention | Quantitative outcomes and findings | Qualitative outcomes and findings |
|---|---|---|---|---|---|---|---|
| 4. | Hahn 2023 [85] Uganda | RCT | **Setting:** Healthcare facility (HIV clinic) **Study population:** adult PLWH. **Sample size:** 269 **Gender:** Male 176 (65.4%) **Age: Mean age**- 50.2 years | **Screening:** AUDIT-C **BI:** MI **Number of sessions:** 2 in-person sessions 3 months apart and several digitally delivered booster sessions in between. **Total duration (both in-person sessions):** 90–140 min **Substance:** alcohol **Delivery:** Ugandan graduates of college and the Uganda Ministry of Health HIV counselors; In person and digital. **Intervention described as brief:** No | Brief advice | **Primary outcome:** number of self-reported drinking days in the prior 21 and biomarker phosphatidyl ethanol (PEth) at six and nine months and viral suppression (<40 copies/mL) at nine months. **Findings:** At follow-up, there were significant reductions in mean number of self-reported drinking days for the live call versus control arm (3.5, 95% CI:2.1–4.9, p < 0.001) and for the technology versus control arm (3.6, 95% CI: 2.2–5.1, p < 0.001). The mean PEth differences compared to the control arm were not significant, i.e. 36.4 ng/mL (95% CI: − 117.5 to 190.3, p = 0.643) for the live call and − 30.9 ng/mL (95% CI: − 194.8 to 132.9, p = 0.711) for the technology arm. Nine-month viral suppression compared to the control was similar in the live call and in the technology arm. **Secondary outcome:** number of heavy drinking days (>3 drinks per day for women and >4 drinks per day for men) in the prior 21 days, the AUDIT-C score (0– 12), unhealthy alcohol consumption (AUDIT-C positive), PEth ≥ 50 ng/mL, a composite measure of unhealthy alcohol use (AUDIT-C positive and/or PEth≥50 ng/mL), referred to as AUDIT-C/PEth, self-reported ART adherence. **Findings:** There were significant decreases in both intervention arms compared to the control at follow up when measured by self-reported number of heavy drinking days, AUDIT-C score, and unhealthy alcohol use by AUDIT-C. There were no significant decreases in the proportion with PEth≥ 50 ng/mL in the intervention arms compared to the control. There were no differences in the intervention arms compared to the control in self-reported prior 30-day ART adherence. | None |
| 5. | L'Engle 2014 [86] Kenya | RCT | **Setting:** HIV prevention drop-in centers **Study population:** female sex workers. **Sample size:** 818 (intervention 410; Control group 408) **Gender:** all female **Age:** Mean age (SD)- 27.5 (6.6) | **Screening:** AUDIT **BI:** MI (model) **Number of sessions:** 6 **Total duration (all sessions combined):** 120 minutes **Substance:** alcohol **Delivery:** Nurses; In person **Intervention described as brief:** yes | 6 nutrition sessions | **Primary outcome:** Difference in AUDIT scores between intervention and control groups **Findings:** There was a statistically significant reduction in alcohol use and binge drinking at 6 and 12 months, with intervention. participants reporting less than one third of the odds of higher levels of drinking than the control group. **Secondary outcome:** Difference in laboratory STI results between intervention and control groups **Findings:** The intervention did not impact laboratory confirmed STI/HIV incidence, self-reported condom use, or sexual violence from nonpaying partners. However, the odds of reporting sexual violence from clients was significantly lower among intervention than control participants at both 6 and 12 months. | None |

(Continued)

**Table 2.** (Continued)

| | Author, year, country | Study design | Study population, Setting sample size; gender/age | Description of intervention | Control intervention | Quantitative outcomes and findings | Qualitative outcomes and findings |
|---|---|---|---|---|---|---|---|
| 6. | Myers 2022 [75] South Africa | RCT | **Setting:** Health facility (primary health care clinics) **Study population:** PLWH of patients with type 1 or type 2 diabetes. **Sample size:** n = 1340 (457 assigned to the dedicated group, 438 assigned to the designated group, and 445 assigned to the treatment as usual group). **Gender:** Female 76% **Age:** Mean age 46 years | **Screening:** AUDIT **BI:** MI and PST **Number of sessions:** 3 with an option of one booster session **Total duration of sessions:** 135 to 180 minutes **Substance:** Alcohol **Delivery:** In person by lay providers **Intervention described as brief:** no | Treatment as usual | **Outcome:** Change in alcohol and depression scores **Findings:** Compared with treatment as usual, the dedicated group (people with HIV adjusted mean difference −5·02 [95% CI −7·51 to −2·54], p<0·0001; people with diabetes −4·20 [−6·68 to −1·72], p<0·0001) and designated group (people with HIV −6·38 [−8·89 to −3·88], p<0·0001; people with diabetes −4·80 [−7·21 to −2·39], p<0·0001) showed greater improvement on depression scores at 12 months. By contrast, reductions in AUDIT scores were similar across study groups, with no intervention effects noted. | None |
| 7. | Parcesepe 2016 [87] Kenya | RCT | **Setting:** HIV prevention drop-in centers **Study population:** adult female sex workers. **Sample size:** 818. **Gender:** all female **Age:** Mean age (SD)– 27.5 (6.6) | **Screening:** AUDIT **BI:** MI (model) **Number of sessions:** 6 **Duration:** 120 min **Substance:** alcohol **Delivery:** Nurse; In person **Intervention described as brief:** yes | 6 session non-alcohol related nutrition intervention. | **Outcome:** effect of intervention on IPV, engagement in sex work, and number of sex partners. **Findings:** The alcohol intervention was associated with statistically significant decreases in physical violence from paying partners at 6 months post-intervention and verbal abuse from paying partners immediately post-intervention and 6-months post-intervention. Those assigned to the alcohol intervention had significantly reduced odds of engaging in sex work immediately post-intervention and 6-months post-intervention. | None |
| 8. | Parry 2023 [88] South Africa | RCT | **Setting:** health facility (HIV clinic) **Study population:** PLWH. **Sample size:** 623. **Gender:** 57.5% female **Age:** mean age 40.8 years [SD = 9.07] | **Screening:** AUDIT **BI:** MI and PST **Number of sessions:** 2 **Total duration (all sessions combined):** 90 min **Substance:** alcohol **Delivery/ In person vs digital:** psychologists/social workers; In person **Intervention described as brief:** yes | Treatment as usual | **Primary outcome:** Number of standard drinks consumed over the past 30 days assessed by questions asked at 6-month follow-up. **Findings:** In support of the hypothesis, an intention-to-treat-analysis for the primary outcome at 6-month follow-up was −0.410 (95% confidence interval = −0.670 to −0.149) units lower on log scale in the intervention group than in the control group (P = 0.002), a 34% relative reduction in the number of drinks. **Secondary outcome:** change in AUDIT score [8], the AUDIT-C score [9] and PEth ng/ml (obtained for 50% of the sample); self-reported ART adherence outcomes. **Findings:** The average number of drinks consumed per month at 3MFU was lower on the log scale, indicating a marginally significant reduction from BL to 3MFU for intervention versus control (P = 0.075). For total AUDIT scores, the decrease from BL to follow-up for the treatment arm was significantly greater than that of the control arm at both 3MFU and 6MFU (P < 0.05) Participants in the intervention group also had lower adjusted log odds of having elevated PEth scores at 3MFU and 6MFU, respec tively (P < 0.05). The decrease in the AUDIT-C score was significantly greater at both time-points in the intervention arm. There were no statistically significant intervention effects for adherence to ART or viral load. | None |

(Continued)

**Table 2.** (Continued)

| | Author, year, country | Study design | Study population, Setting sample size; gender/age | Description of intervention | Control intervention | Quantitative outcomes and findings | Qualitative outcomes and findings |
|---|---|---|---|---|---|---|---|
| 9. | Peltzer 2013 [89] South Africa | RCT | **Setting:** health facility (primary care) **Study population:** patients with TB. **Sample size:** 1120. **Gender:** Male- 74.3% **Age:** averaged 36.7 years of | **Screening:** AUDIT **BI:** Education and MI (Information-Motivation-Behavioral Skills Model) **Number of sessions:** 2 **Total duration (all sessions combined):** 30–40 min **Substance:** alcohol **Delivery:** HIV counselors; In-person **Intervention described as brief:** yes | Health education leaflet on responsible drinking. | **Primary Outcome:** Change in the mean AUDIT score between baseline and month 3. **Findings:** The intervention effect on the AUDIT score was statistically not significant. **Secondary outcome:** successful TB response, classified by WHO as cured or treatment completed (versus treatment failure, defaulted, died or transferred out to another health facility **Findings:** At 6-month follow-up the intervention group did not significantly differ to the control group in terms of TB treatment cure or completion. | None |
| 10. | Puryear 2023 [90] Uganda, Kenya | RCT | **Setting:** healthcare facilities (HIV clinics); alcohol-serving venues **Study population:** PLWH. **Sample size:** 401 persons (198 intervention, 203 control) **Gender:** Female- 131 (33%) **Age:** Median (IQR)- 37 [31,43] | **Screening:** AUDIT-C **BI:** Education and MI (Information-Motivation-Behavioral Skills Model) **Number of sessions:** two counseling sessions, with brief "booster" phone-based counseling session **Total duration (all sessions combined):** not described **Substance:** alcohol **Delivery:** Lay counselors; In-person and phone-based **Intervention described as brief:** yes | Brief advice on the harmful effects of alcohol and safe levels of drinking | **Primary outcome:** viral suppression 24 weeks after enrolment. **Findings:** Viral suppression did not differ between arms at 24 weeks: suppression was 83% in intervention and 82% in control arms (RR: 1.01, 95% CI: 0.93–1.1). Among PLWH with baseline viral non-suppression, 24-week suppression was 73% in intervention and 64% in control arms (RR 1.15, 95% CI: 0.93–1.43). **Secondary outcome:** changes in unhealthy alcohol use at 24 weeks, assessed by AUDIT-C and phosphatidylethanol (PEth), an alcohol biomarker. **Findings:** Unhealthy alcohol use declined from 98% at baseline to 73% in intervention and 84% in control arms at 24 weeks (RR: 0.86, 95% CI: 0.79–0.94). | None |
| 11. | Sony 2007 [91] Sudan | RCT | **Setting:** Health facility (TB clinics) **Study population:** Males, enrolled for PTB treatment, and staff (medical assistants) **Sample size:** staff -48 patients- 513 **Gender:** Males only **Age:** not provided | **Screening:** Single question asking about tobacco use **BI:** MI **Number of sessions:** 4 **Total duration (all sessions combined):** Not provided. **Substance:** tobacco **Delivery:** Medical assistants; In-person **Intervention described as brief:** yes | None | **Outcome:** Acceptability of intervention by staff **Findings:** Overall, the number of medical assistants who reported discussing tobacco 'almost always' increased from 22 to 29 (60%). **Outcome:** TB treatment outcomes **Findings:** Enrolled patients had higher cure rates, lower default rates, and lower rates of transfer compared to non-enrolled patients. The differences were not significant for death rates or failures. **Outcome:** Change in tobacco use between baseline and 12 months follow up). **Findings:** 66% of those who completed treatment reported no longer using any form of tobacco at the end of the trial. 53.6% of 308 patients in the intervention group reported quitting vs. 14.3% of the 42 patients in the control group, although the difference was not statistically significant. | |

(Continued)

**Table 2.** (Continued)

| | Author, year, country | Study design | Study population, Setting sample size; gender/age | Description of intervention | Control intervention | Quantitative outcomes and findings | Qualitative outcomes and findings |
|---|---|---|---|---|---|---|---|
| 12. | Sorsdahl 2021 [92] South Africa | RCT | **Setting:** Community **Study population:** adults. **Sample size:** 60 (intervention 30, control 30) **Gender:** Male 32 (53.3%) **Age:** range- 18–65; Mean age—31.0 (6.5) | **Screening:** **BI:** motivational enhancement therapy, exposure therapy, behavioral therapy, relapse prevention, cognitive therapy, family therapy **Number of sessions:** 7 **Total duration (all sessions combined):** not provided **Substance:** methamphetamine **Delivery:** Clinical psychologists; In-person **Intervention described as brief:** yes | Treatment as usual | **Primary outcome:** reductions in frequency of methamphetamine use at week 6 and month 3. **Findings:** In the intention-to-treat analysis, the intervention group displayed greater reductions in frequency of methamphetamine use than the control group at week 6 and month 3. **Secondary outcome:** Change in depression, anxiety, cravings and disability scores, **Findings:** The intervention group reported significantly lower anxiety scores than the control group (p = 0.049) at both 6 weeks and at 3 months post-intervention. There were no significant differences between groups for the other secondary out-come measures. **Outcome:** Session attendance **Findings:** Of the 30 participants who were randomized to the intervention group, 26 completed session one, 24 completed two sessions, 21 completed three sessions, 18 completed four sessions, 15 completed five sessions and 13 (43%) completed all six sessions. | None |
| 13. | Staton 2022 [93] Tanzania | RCT | **Setting:** Health facility (Emergency department) **Study population:** patients with acute injuries, with AUDIT > 8 **Sample size:** 41 (23 in standard booster and 18 in personalize booster arm) **Gender:** >96% males **Age:** Mean age (SD) 37 years (15) | **Screening:** AUDIT **BI:** MI **Number of sessions:** one 15 min MI session followed by weekly SMS booster—either standard or with personalized content over 3 months. **Substance:** Alcohol **Delivery:** Nurse; In-person, and digital **Intervention described as brief:** Yes | Usual care- normal discharge instructions | **Primary outcome:** feasibility of using SMS booster to augment a BI program. **Findings:** Follow up at 6 months- 90% of those in standard booster and 81 in personalized messages remained in treatment. Out of 41, 38 received the messages correctly. 52% of expected messages were attempted to be sent. Majority of those not sent was due to system error. Only 34% of messages were delivered during the intended period; 33% of messages had failed delivery. Reasons for failed delivery were participants traveling far and phone switched off. 89% received treatment as allocated. **Secondary outcome:** Acceptability **Findings:** At 6 months follow up- 85% of eligible participants agreed to be interviewed 91% reported large positive effect 90% reported satisfaction with the timing due to the opportunity to read the messages. 10% could not remember about the intervention. Only 4% opted to stop receiving the sms booster over the 6 months period. | None |
| 14. | Washio 2020 [74] South Africa | Reports part of data from a larger RCT | **Setting:** Hospital **Study population:** pregnant women 18–45 years, reporting at least 1 drug use in past month. **Sample size:** 33 **Gender:** all female **Age:** mean age 30 years | **Screening:** Questions on recent drug use and breathalyzer alcohol test **BI:** Women health coop (WHC) intervention. based on knowledge and skills to reduce drug use, sex risk and violence with role play and rehearsal **Number of sessions:** 4 modules delivered in 2 sessions in groups of 4–6 women. **Total duration for all sessions:** 2 hours (120 min) **Substance:** multiple substances **Delivery:** Peer educator; In-person **Intervention described as brief:** yes | None | **Primary outcome:** ART adherence **Findings:** improved from 80% at baseline to 100% at follow up. **Secondary outcome:** change in alcohol use **Findings:** All 5 participants were at risk of alcohol use at baseline, but only 2 were at risk during follow-up assessment. No positive breathalyzer report at follow-up. 80% completed the WHC program. 20% had physical abuse at baseline, but none during follow up. overall reduction in number of alcohol use days in past one month reduced from12 to average of 4.3. | None |

*(Continued)*

**Table 2.** (Continued)

| | Author, year, country | Study design | Study population, Setting sample size; gender/age | Description of intervention | Control intervention | Quantitative outcomes and findings | Qualitative outcomes and findings |
|---|---|---|---|---|---|---|---|
| 15. | Wechsberg 2013 [94] South Africa | RCT | **Setting:** Community **Study population:** Women of childbearing age (18–33), who used at least 2 drugs weekly in past 3 months. **Sample size:** 720. **Gender:** All Female **Age:** Mean age (SD)- 23.2 years (4.2) | **Screening:** Questions on recent drug use and breathalyzer alcohol test **BI:** Women health coop (WHC) intervention based on knowledge and skills to reduce drug use, sex risk and violence with role play and rehearsal **Number of sessions:** 4 modules delivered in 2 sessions in groups of 4–6 women. **Total duration for all sessions:** 2 hours (120 min) **Substance:** multiple substances **Delivery:** Peer educator; In-person **Intervention described as brief:** yes | 2 control arms **nutrition intervention** - received information on healthy food and food preparation (equal attention as the WHC arm) **HIV Counseling and Testing (HCT):** Standard HIV pre-test and posttest counselling only | **Primary outcome:** **Effectiveness**- drug use; biologically confirmed abstinence. **Findings:** At 12 months, more abstinence in intervention arm (26.9%) compared to nutrition intervention arm (16.9%) and HCT arm (20%). Overall, changes observed in the proportion of drug use was higher in the WHC arm than both nutrition and HCT arms. **Secondary outcome:** impaired sex due to substance use; intervention feasibility. **Findings:** lower sexual impairment in WHC arm. Higher proportion reporting to be sober during sexual interaction in WHC arm than nutrition intervention arm (65.9% versus 54.4%, p<0.001). Feasibility- Overall retention 81.7% (82% intervention and 83% control). Average follow up at 12 months 85% | None |
| 16. | Zule 2014 [95] South Africa | RCT | **Setting:** Community **Study population: PLWH** (Females between 18 and 33 years) **Sample size:** n = 84 **Gender:** All female **Age:** Intervention group 23.9 (SD 4.2) Comparison groups 22.8 (SD = 3.7) | **Screening:** Questions on recent drug use and breathalyzer alcohol test **BI:** Women health coop (WHC) intervention. based on knowledge and skills to reduce drug use, sex risk and violence with role play and rehearsal **Number of sessions:** 4 modules delivered in 2 sessions in groups of 4–6 women. **Total duration for all sessions:** 2 hours (120 min) **Substance:** multiple substances **Delivery:** Peer educator; In-person **Intervention described as brief:** yes | 2 control arms **nutrition intervention** - received information on healthy food and food preparation (equal attention as the WHC arm) **HIV Counseling and Testing (HCT):** Standard HIV pre-test and posttest counselling only | **Primary outcome:** Alcohol use at follow up; Positive urine drug screen at follow up **Findings:** women in the WHC intervention condition were more likely to report that they were abstinent from alcohol than women assigned to the comparison conditions (OR = 3.61; 95% CI = 1.23, 11.70; p = 0.016. Women assigned to the WHC intervention condition were somewhat more likely to provide a urine sample that tested negative for all four drugs (OR = 3.07; 95% CI = 0.83, 12.31; p = 0.105) than women in the combined comparison group | None |
| **Quasi-experimental studies** | | | | | | | |
| 17. | Calligaro 2021 [76] South Africa | Quasi-experimental | **Setting:** Health facility **Study population:** Adult patients with multi-drug resistance TB **Sample size:** n = 39 **Gender:** Male 54% **Age:** Median age 36 years | **Screening:** ASSIST **BI:** MI and relapse prevention **Number of sessions:** 4 **Total duration:** 3-4hr **Substance:** multiple **Delivery:** Counsellors with at least a Bachelor qualification in either psychology or social work; In-person **Intervention described as brief:** Yes | None | **Outcome:** Change in scores for substance use (ASSIST), depression, psychological distress, functional impairment, nicotine dependence, readiness for change and perceived social support. **Findings:** The ASSIST score decreased following the intervention (pre-intervention median (IQR) 17.5 (15–24) versus 6 (6–8)). Depressive symptomatology improved, with the median (IQR) CES-D score significantly lower post intervention [23.5 (18–34) vs. 17 (13–22)]. There were also significant improvements in nicotine dependence, health status, functional impairment, psychological distress, readiness for change and perceived social support. | **Outcome:** Patients's experiences with the intervention **Findings:** High levels of motivation for addressing substance use were reported by all patients. Patients reported positive experiences with the program. There was consensus that the content, structure, and delivery of the program were acceptable. Patients could not identify any aspect that was redundant or not applicable to their needs. |

*(Continued)*

**Table 2.** (Continued)

| | Author, year, country | Study design | Study population, Setting sample size; gender/age | Description of intervention | Control intervention | Quantitative outcomes and findings | Qualitative outcomes and findings |
|---|---|---|---|---|---|---|---|
| 18. | Odukoya 2014 [77] Nigeria | Quasi-experimental | **Setting:** Secondary schools **Study population:** Secondary school students. **Sample size:** 973 (478 intervention group; 495 control group). **Gender:** Male- 230(48.1%) **Age:** Mean age SD; 14.08 (2.09) | **Screening:** none **BI:** health education grounded on health belief model **Number of sessions:** 2 health talks **Total duration (all sessions combined):** not provided **Substance:** tobacco **Delivery: Who?:** In-person group based. **Intervention described as brief:** yes | No intervention | **Primary outcome:** Change in knowledge scores of harmful effects of cigarettes; Change in mean anti-smoking attitudinal scores; Proportion of students (never-smokers) who report that they will initiate smoking within the next twelve months; · Self-reported tobacco use in the past 30 days. **Findings:** Students in the intervention group had significantly higher mean knowledge scores after the program (8.39 ±3.4vs.11.98±3.1; p<0.001). p<0.001). The number of students who reported that they had smoked in the last 30 days decreased by 1.0% in the intervention group (4.0% vs. 3.0% p = 0.413) but increased by 0.5% in the control group (3.5% vs. 3.5% p = 0.661). These differences were, however, not statistically significant. The proportion of current smokers who desired to quit increased significantly from 47.4% to 85.7% (p = 0.029) after the intervention and this difference was statistically significant. The proportion of never-smokers who reported that they would initiate smoking within the next year reduced from 18.7% to 12.7% post intervention. (p = 0.028). | None |
| 19. | Takahashi 2018 [96] Kenya | Quasi-experimental | **Setting:** Community **Study population:** Adults 18–65 years, with an AUDIT score of 8–19. **Sample size:** 161 (52 in the control group, 52 in the only ABI group and 57 in the ABI + MT group.) **Mean age** (SD): Control 40.8 (1.7) ABI only 46.4(1.6) ABI+MT 44.7(1.7) **Gender:** Male 90.4%, | **Screening:** AUDIT **Intervention 1:** **BI:** Brief MI (model) **Number of sessions/duration:** Three sessions, each lasting 5–20 min. **Total duration (all sessions combined):** 15-60min. **Substance:** Alcohol **Delivery:** Lay provider; In-person **Intervention 2:** BI: Motivational talks plus MI (3 sessions); delivered by lay providers and persons in recovery **Intervention described as brief:** Yes. | **Control:** General health information on alcohol use. **Second arm:** Motivational talks plus BI (3 sessions). | **Outcomes:** difference in AUDIT mean scores between the control and intervention arms. **Findings:** Unadjusted mean AUDIT scores were reduced in all three study groups between the baseline and 6-month assessment post-intervention, with reductions from 13.4 to 9.4, 14.4 to 7.7 and 14.1 to 6.6 in the control, only ABI and ABI + MT groups, respectively. Over time, there was a greater reduction in the adjusted mean AUDIT scores in the intervention groups than in the control group, with the greatest reduction being observed in the ABI + MT group. | None |

**Mixed methods**

| | Author, year, country | Study design | Study population, Setting sample size; gender/age | Description of intervention | Control intervention | Quantitative outcomes and findings | Qualitative outcomes and findings |
|---|---|---|---|---|---|---|---|
| 20. | Asombang 2022 [97] Zambia | Mixed methods | **Setting:** Health facility (HIV clinics) **Study population:** PLWH (Adults) **Sample size:** n = 693 **Gender:** Male 45% **Age:** Median age- 34 years | **Screening:** AUDIT-C **BI:** brief and unstructured information on reducing alcohol use **Number of sessions:** one brief talk on alcohol use and health talks given over an unknown number of times, on harms of alcohol use. **Duration:** Not indicated **Substance:** Alcohol **Delivery:** Counselors; In-person **Intervention described as brief:** Yes | No intervention | **Primary outcome:** Change in alcohol use between baseline and 12 months **Findings:** During the first year on ART the overall prevalence of unhealthy alcohol use in the analysis group reduced significantly from 40.4% to 29.6% (P<0.01). Of the 280 with unhealthy use at baseline, 122 (43.6%) reported a lower degree of alcohol consumption at 1 year. Reduction from unhealthy to moderate use or abstinence was more common with older age, female, and non-smoking (all P<0.05) | **Outcome:** perspectives, beliefs, and norms surrounding alcohol consumption in relation to HIV infection and care received at the clinic we **Findings:** Participants reported ineffective alcohol support at clinics, social pressures in the community to consume alcohol, and unaddressed drivers of alcohol use including poverty, poor health status, depression, and HIV stigma. |

*(Continued)*

**Table 2.** (Continued)

| | Author, year, country | Study design | Study population, Setting sample size; gender/age | Description of intervention | Control intervention | Quantitative outcomes and findings | Qualitative outcomes and findings |
|---|---|---|---|---|---|---|---|
| 21. | Carney 2020 [98] South Africa | Mixed methods | **Setting:** Community **Quantitative** **Study population:** Adolescents. **Sample size:** n = 30 adolescents and caregivers **Gender:** Adolescents: Male (56.7%) Caregivers: 2 (6.7%) **Age:** Adolescents: Mean (SD) 15.5 years (1.4) Caregivers: not provided **Qualitative** **Sample size:** n = 20 (10 Adolescents and 10 caregivers). | **Screening:** questions asking about monthly substance use in the past 6 months **BI:** MI and CBT **Number of sessions:** 3 (2 for adolescent and 1 for caregiver) **Duration:** not provided **Substance:** multiple **Delivery:** Psychological counselor; In person **Intervention described as brief:** Yes | None | **Primary outcome:** Change in levels of substance use **Findings:** For those participants who reported current use of alcohol, there was a significant reduction in the self-reported fre-quency of alcohol use in the previous 30 days (t(14) = 2.3, p = .04, Cohen's d = 0.58) and the number of drinks containing alcohol in one sitting (t(14) = 2.7, p = .02, Cohen's d = 0.69) from baseline to the follow-up appointment. Self-reported cannabis use frequency in the previous 30 days also decreased significantly, (t(19) = 2.1, p = .05, Cohen's d = 0.47), but changes in in the proportion of adolescents with positive biological results for cannabis did not reach statistical significance (X2(1) = 4.9, p = .06). **Secondary outcome:** Change in sexual and delinquent behavior **Findings:** There was a significant reduction in self-reported delinquent-type behaviors from baseline to the follow-up appointment (t(27) = 2.9, p < .01, Cohen's d = 0.55). Sexual risk behavior decreased, although the difference did not reach statistical significance (t(27) 2.0, p = .06, Cohen's d = 0.38). | Participants experienced the intervention as a safe haven, and as leading to behavior change. They also expressed the importance of extending the reach of the program and ensuring its sustainability due to high levels of substance use among adolescents in the community. |
| 22. | Louwagie 2019 [99] South Africa | Mixed methods | **Setting:** Healthcare facility **Study population:** Adult patients with TB **Sample size:** 45 **Gender: Male-** 82.2% **Age:** Mean age- 39.8 years | **Screening:** AUDIT **BI:** Education and MI **Number of sessions:** 3 MI in-person sessions and SMSs with education on tobacco and alcohol harms, and TB medication adherence. **Total duration (all sessions combined):** 60 min for in-person sessions. **Substance:** tobacco, alcohol **Delivery:** Lay providers; In-person and digital (SMS) **Intervention described as brief:** Yes | None | **Primary outcome:** Intervention feasibility **Findings:** Nearly one third of the patients were eligible because of exclusively drinking (28.9%), 31.1% because of exclusively smoking and 40.0% for concurrently drinking and smoking. Nearly all patients (93.3%) received their first MI counseling session, but close to 30% failed to receive their second or third MI session due to various reasons. Two patients died due to causes that were unrelated to the study. | **Outcome:** Patient perceptions about intervention **Findings:** Patients reported the intervention was helpful, enjoyable, and helped them reduce smoking and drinking. Most patients reported better adherence to TB treatment. All patients reported liking their interaction with counselor. Several patients did not like the session being recorded, some struggled to understand messages or had technical issues. **Outcome:** Perceptions of healthcare workers on intervention: **Findings:** Overall, lay providers felt positive about the training received; They however felt unprepared for the diversity of problems, attitudes, and beliefs that patients presented with. |
| 23. | Myers 2018 [78] South Africa | Mixed methods | **Setting:** Health facility (Primary health clinics) **Study population:** Chronic disease patients **Sample size:** n = 40 **Gender:** 85% female **Age:** Mean (SD) 44 years (12.2) Community Health workers n = 7 | **Screening:** AUDIT **BI:** MI and PST **Number of sessions:** 3 **Duration:** Not indicated **Substance:** Alcohol **Delivery:** lay-providers; In person **Intervention described as brief:** No | None | **Outcome:** Feasibility of recruitment and retention **Findings:** Of the 553 chronic disease patients screened for recent alcohol use and depressed mood, 262 (48%) were potentially eligible for study inclusion and referred for eligibility screening. About a quarter (26%, n = 69) declined screening, mainly due to lack of time or interest. Of the 193 remaining patients, 101 met inclusion criteria. Sixteen patients (16%) were eligible on their AUDIT scores, 69 (58%) on their CES-D scores and 16 (16%) on their AUDIT and CES-D scores. Sixty-seven (66%) of these eligible patients were interested in participation. Only 40 of these 67 patients returned for their enrolment visit; the remainder were untraceable. **Secondary outcome:** Feasibility **Findings:** The primary outcomes supported the feasibility of a larger study. | **Outcome:** Healthcare worker perceptions on the intervention **Findings:** All CHWs viewed the intervention as acceptable and beneficial to patients with chronic disease |

(*Continued*)

**Table 2.** (Continued)

| # | Author, year, country | Study design | Study population, Setting sample size; gender/age | Description of intervention | Control intervention | Quantitative outcomes and findings | Qualitative outcomes and findings |
|---|---|---|---|---|---|---|---|
| 24. | O'Grady 2022 [100] Mozambique | Mixed methods | **Setting:** Community **Study population:** Psychiatric technicians and primary care providers. **Qualitative respondents: Sample size:** 15 **Age:** mean age (SD) 28.66 (7.01). **Gender:** Male n = 13 **Quantitative respondents: Sample size:** n = 45 **Age:** Mean age (SD) 30.57 (6.08) **Gender:** majority were women. | **Screening:** AUDIT **BI:** brief MI **Number of sessions:** 4 **Duration:** 150 min **Substance:** Alcohol **Delivery:** provider facing mobile health application **Intervention described as brief:** Yes | None | **Outcome:** Acceptability of Intervention Measure, Intervention Appropriateness Measure, Feasibility of Intervention Measure **Findings:** app acceptability (scale mean = 4.33/5.0); app appropriateness (scale mean = 4.22/5); app feasibility vb (scale mean = 4.17/5). | **Outcome:** app acceptability, feasibility, appropriateness **Findings:** the HCWs felt that the app provided for an efficient session because of the automated calculation of the screening tool and its structured resources; that the app provided a good user experience; that the app content helped with patient interactions; that the app was a good match for them to do the BI |
| 25. | van der Westhuizen 2021 [79] South Africa | Mixed methods | **Setting:** Health facility **Study population:** patients with moderate to high-risk substance use. **Sample size:** Quantitative n = 4847 Qualitative n = 18 **Gender:** Males 74% for quantitative sample and 50% male for the qualitative interview sample **Age:** 65% below 35 years | **Screening:** ASSIST **BI:** MI plus PST **Number of sessions:** 3 **Duration:** not specified **Substances:** Multiple **Delivery:** Facility-based counsellors; In-person **Intervention described as brief:** yes | | **Primary outcome:** Substance use outcomes. **Findings:** For those with alcohol as primary substance, the number of alcohol use days were reduced significantly between baseline and 3 months. e.g. number of days with no alcohol use reduced from 11% at baseline to 59% at 3 month follow up. For other drugs, a similar pattern was observed with a drop in number of drug use days. **Secondary outcome: Patient satisfaction-** 97% said they would return for the service; 98% agreed that program should be provided to other settings. **Acceptability-** many participants found the program to be helpful. | **Outcome:** Acceptability of program **Findings:** Patients reported that the counselors were non-judgmental and caring; **Perceived benefits included:** reduced substance use, through implementing the strategies learnt; improvement in health e.g. less injuries; improved interpersonal relationships; ability to save money due to less substance use. |
| 26. | van der Westhuizen 2019 [101] South Africa | Mixed methods | **Setting:** Health facility; community **Study population:** Stakeholders at different levels e.g. hospital, regional and policy level. **Sample size:** 40 **Gender:** 74% males **Age:** Mean age 33 years | **Screening:** ASSIST **BI:** MI plus PST **Number of sessions:** 3 **Duration:** not specified **Substances:** Multiple **Delivery:** Facility-based counsellors; In-person **Intervention described as brief:** yes | None | **Outcome:** Intervention feasibility **Findings:** 37% of those screened met the criteria for risky substance use on ASSIST. 83% received the 1st session at the acute visit. | **Outcome:** Intervention acceptability **Findings:** Stakeholders agreed that the program met a need in community especially due to high burden of injuries related to substance use. **Outcome:** Appropriateness **Findings:** Most of the healthcare workers felt that the program did not interfere with the acute care of patients. |
| **Qualitative studies** | | | | | | | |
| 27. | Gichane 2023 [102] Uganda | Qualitative | **Setting:** Health facility (HIV clinic) **Study population:** PLWH. **Sample size:** n = 36 **Gender and age:** not provided | **Screening:** AUDIT-C **BI:** MI **Number of sessions:** 2 in-person sessions 3 months apart and several digitally delivered booster sessions in between. **Total duration (both in-person sessions):** 90–140 min **Substance:** alcohol **Delivery:** Ugandan graduates of college and the Uganda Ministry of Health HIV counselors; In person and digital. **Intervention described as brief:** yes | Brief advice | None | **Outcome:** Participant perceptions about the intervention **Findings:** Participants reported reduced alcohol use, increased financial savings, awareness of the social and physical impact of alcohol. Peer pressure, and lack of social support constituted barriers to alcohol reduction. |

(*Continued*)

**Table 2.** (Continued)

| Author, year, country | Study design | Study population, Setting sample size; gender/age | Description of intervention | Control intervention | Quantitative outcomes and findings | Qualitative outcomes and findings |
|---|---|---|---|---|---|---|
| 28 | Myers 2020 [81] South Africa | Qualitative | **Setting:** Health facility (HIV clinics) **Study population:** PLWH. **Sample size:** n = 49 **Gender:** Male 33% **Age:** Mean age 41 years | **Screening:** AUDIT **BI:** MI and PST **Number of sessions:** 4 **Duration:** not specified **Substance:** alcohol **Delivery:** In person by trained counselors **Intervention described as brief:** yes | None | None | **Outcome:** Perceptions of participants on intervention. **Findings:** Participants believed that it was acceptable to offer PLWH, an alcohol reduction intervention during HIV treatment. |
| 29 | Myers 2017 [80] South Africa | Qualitative | **Setting:** Health facility (HIV clinics) **Study population:** PLWH. **Sample size:** 11 **Gender:** Male - 8 **Age: not provided** | **Screening:** AUDIT **BI:** MI and PST **Number of sessions:** 4 **Duration:** not specified **Substance:** alcohol **Delivery:** In person by HIV counselors. **Intervention described as brief:** yes. | None | None | **Outcome:** Perceptions of participants acceptability of the intervention. **Findings:** The MI-PST intervention was highly acceptable to participants. As a result of the intervention, participants reported less use of alcohol as a coping mechanism. |
| 30. | Williams 2020 [103] South Africa | Qualitative | **Setting:** Health facility (primary health care setting) **Study population:** PLWH (young women) **Sample size:** 27 **Gender:** All females **Age:** 18–35 years, mean 28 years | **Screening:** AUDIT **BI:** MI **Number of sessions:** 4 (initial 3 sessions delivered over 6 weeks and an optional 4th session) **Duration:** 6 weeks and optional 2 weeks Follow u at 6 and 12 months **Substance:** Alcohol **Delivery/** In person vs digital: In-person by lay counselors **Intervention described as brief:** Yes | None | **None** | **Outcome:** Participants' experiences with the intervention. **Findings:** The participants reported that the intervention led to improved knowledge and health- promoting behavior change; They also reported improvement in thought patterns and attitudes towards life circumstances. The participants appreciated the quality of relationship with the counselors and felt that the content and delivery of intervention was appropriate. |

**Abbreviations:** ABI- Alcohol Brief Interventions, AIDS- Acquired Immunodeficiency Syndrome, ART- Antiretroviral Therapy, ASSIST- Alcohol, Smoking and Substance Involvement Screening Test, AUDIT- Alcohol Use Disorders Identification Test, AUDIT–C- Alcohol Use Disorders Identification Test-Concise, BI- Brief Intervention, BL- Baseline, CBT- Cognitive Behavioral Therapy, CES-D- Center for Epidemiologic Studies Depression Scale, CHWs- Community Health Workers, FGD- Focus Group Discussions, HCWs- Healthcare Workers, HIV- Human Immunodeficiency Virus, IPV-Interpersonal Violence, IQR- Interquartile range, MFU-Month Follow Up, MI- Motivational Interviewing, MT-Motivational Talks, PLWH- People Living With Human Immunodeficiency Virus, PST- Problem Solving Therapy, PTB- Pulmonary tuberculosis, RCT- Randomized Control Trial, SD- Standard Deviation, SMS-Short Message Service, STIs- Sexually Transmitted Infections, TB- Tuberculosis, WHO- World Health Organization.

**Table 3. Feasibility of BIs with session numbers not described.**

| | Author, year, country | Study design | Study population, Setting sample size; gender/age | Description of intervention | Control intervention | Quantitative outcomes and findings | Qualitative outcomes and findings |
|---|---|---|---|---|---|---|---|
| 1. | Sorsdahl 2013 [107] South Africa | Qualitative | **Setting:** Health facility (emergency department) **Study population:** Healthcare workers. **Sample size:** n = 24 **Gender:** Female 83% **Age:** Mean age (SD)- 37 years (12.4). | The article talks broadly about SBI with intervention details not provided. | None | None | **Outcome:** Healthcare workers perceptions on SBIs guided by CFIR. **Findings:** All participants perceived substance use to be highly prevalent amongst members of the community. Respondents were concerned that current efforts to address substance use amongst patients presenting for ED services were inadequate. Barriers to implementation included: aggressive patients, lack of private space within the ED, severely injured patients |
| 2. | Williams 2015 [108] South Africa | Qualitative | **Setting:** Health facility (midwife obstetric units) **Study population:** Healthcare workers. **Sample size:** n= 43 **Gender:** All female **Age:** Mean age (SD) 41.9 (9.3) | The article talks broadly about SBI with intervention details not provided. | None | **None** | **Outcome:** Midwives' perceptions on substance use and available interventions. **Findings:** substance use in the community was seen as being widespread and alcohol and other drugs were easily accessible. Most participants were very confident that they have sufficient knowledge to identify alcohol and other drug use in the women to whom they provide services. There was no formal protocol to address substance use issues in the facility. |
| 3. | Miller 2021 [104] Uganda | Qualitative | **Setting:** Health facility **Study population:** PLWH (women) **Sample size:** n = 20 **Gender:** All female **Age:** Mean age 31 years | The article talks broadly about SBI with intervention details not provided. | None | None | **Outcome**: Participant perceptions on the intervention **Findings:** Most participants felt an alcohol SBI was both acceptable and needed. Some expressed skepticism that participation in an intervention would effectively prompt behavior change, despite the harms of heavy alcohol use. |

(*Continued*)

**Table 3.** (Continued)

| | Author, year, country | Study design | Study population, Setting sample size; gender/age | Description of intervention | Control intervention | Quantitative outcomes and findings | Qualitative outcomes and findings |
|---|---|---|---|---|---|---|---|
| 4. | Musyoka 2023 [106] Kenya | Quasi-experimental | **Setting:** College **Study population:** undergraduate first-year student peer mentors aged between 18 years to 25 years. **Sample size:** n = 100 (51 experimental, 49 control) **Gender:** Males 61.9% **Age:** Mean age 19 years | **Screening**: AUDIT and ASSIST **BI:** the intervention was either a brief counselling session, a scheduled follow-up session, and/or referral for further counselling by the campus student counsellor. **Details of session content, number of sessions, duration of sessions:** not provided **Substance:** multiple **Delivery:** In-person guided by an mhealth app in treatment arm **Intervention described as brief:** yes | Peer mentors delivered BI using a standard paper-and-pen-based tool | **Primary outcome:** Intervention acceptability, appropriateness, feasibility, and reach **Findings:** The mHealth-based peer mentoring tool scored high with 100% of users rating it as feasible and acceptable. Among the two study cohorts, there were no differences in the acceptability of the peer mentoring intervention. | None |
| 5. | Pascall 2022 [105] South Africa | Cross-sectional | **Setting:** Community **Study population:** adult **Sample size:** 1294 **Gender:** Male 70% **Age:** Mean age (SD) 32.7 (10.6) | **Screening:** single question about alcohol use **BI:** advice to reduce or stop drinking alcohol **Number of sessions:** not described. **Duration:** not described **Substance:** alcohol **Delivery:** Healthcare worker; In person vs digital: **Intervention described as brief:** yes | None | **Outcome:** implementation of SBI in primary health care **Findings:** Among drinkers at risk for AUD, alcohol use screening rates were 6.7%; and brief intervention rates were 4.6% | none |
| 6. | Peltzer 2008 [34] | Cross-sectional | **Setting:** Churches **Study population:** Clergy **Sample size:** n = 117 **Gender:** 73% Male **Age:** Mean (SD) 44.2 years (11.6) | The article talks broadly about SBI with intervention details not provided. | None | **Outcome:** Frequency and type of screening and BIs clergy use for substance problems in members of their congregation **Findings:** Four in five clergy offered advice and spiritual solace when approached about alcohol or drug abuse problems; 40% of clergy referred congregants with substance use problems to a social worker, therapist, treatment program or self-help group. | None |

**Abbreviations:** ASSIST- Alcohol, Smoking and Substance Involvement Screening Test; AUD- Alcohol Use Disorder; AUDIT- Alcohol Use Disorders Identification Test; BI- Brief Intervention; CFIR-Consolidated Framework for Implementation Research; ED- Emergency Department; PLWH- People Living with Human Immunodeficiency Virus; SBIs- Screening and Brief Interventions; SD- Standard Deviation.

emphasizing client responsibility and asking for permission to screen and give feedback [42]), understanding, lack of support from within the facility, and poor continuity of care [110] (Table 4).

**Table 4. Summary of studies on feasibility and effectiveness of BI provider training.**

| | Author, year, country | Study design | Study population, Setting sample size; gender/age | Description of brief intervention | Training details | Quantitative outcomes and findings | Qualitative outcomes and findings |
|---|---|---|---|---|---|---|---|
| 1. | Peltzer 2006 [112] South Africa | Quasi-experimental | **Setting:** Health care facilities **Study population and sample size:** n = 236 (121 nurses, 86 professional nurses, 19 enrolled nurses, and 10 assistant nurses) **Age:** Mean age 39.8 years **Gender:** Females 90.9% | Screening with AUDIT followed by MI based on model. **Intervention recipients:** patients attending primary health care clinics | **Mode of training:** Face to face by a nurse and psychologist trainer **Duration of training:** 6 hours | **Outcomes:** Effectiveness of training assessed by quantitatively evaluating variables below: Objective knowledge, Confidence in screening, Confidence in brief intervention, Perceived obstacles to screening, Perceived obstacles to brief intervention, Self-efficacy in SBI, expectations of SBI benefit. **Findings:** Change in scores was statistically significant for these variables: Objective knowledge Confidence in screening Confidence in brief intervention, self-efficacy in SBI. | NA |
| 2. | Odukoya 2020 [111] Nigeria | Quasi-experimental | **Setting:** Health facility **Study population:** Physicians. **Sample Size:** 946 **Gender:** Male- 53.3% **Age:** Mean age- 37.7 ± 7.4 years. | **Screening:** not specified **BI:** Ask, Advise and Refer (AAR) model **Duration:** 2 min **Substance:** tobacco **Delivery:** Physicians, in person **Intervention described as brief:** yes **Intervention recipients:** Patients 12 years and older seeking care at tertiary level hospitals | **Mode of training:** Weekly text messages reminding physicians to conduct the BI (for 13 weeks) | **Primary outcomes:** Awareness of the AAR based BI; proportion of physicians who offered each of the AAR components to at least 50% of eligible patients in past 3 months. **Findings:** Awareness of the AAR increased significantly by 58.5% after the intervention, whereas inquiry of tobacco status increased by 25.0%. Participants who reported advising and referring patients who smoked also increased significantly, but by slightly lower percentages, i.e., 19.8% and 12.3%, respectively. **Secondary outcomes included:** Self-reported effects of the messages on motivation to offer AAR; physicians' attitudes towards the messages. **Findings:** 71.5% of respondents who received the messages admitted to having read them, either all the time or most times. 72.7% acknowledged that the messages were helpful in increasing their knowledge; 87.9% reported that the messages had motivated them to practice AAR. A majority (84.8%) felt that the frequency of the messages was "just adequate." | |

*(Continued)*

**Table 4.** (Continued)

| | Author, year, country | Study design | Study population, Setting sample size; gender/age | Description of brief intervention | Training details | Quantitative outcomes and findings | Qualitative outcomes and findings |
|---|---|---|---|---|---|---|---|
| 3. | Malan 2016 [109] South Africa | Quasi-experimental | **Setting:** Health facility<br><br>**Study population:** Health care workers (23 nurses, 12 family medicine registrars, 2 general practitioner doctors and 4 family physicians).<br><br>**Sample size:** 41.<br><br>**Gender:** Females 82%<br>**Age:** not indicated<br><br>**Mode of training:** The study participants were provided with recordings on integrated two behavior change approaches, the 5 As and a guiding style derived from motivational interviewing. The training was designed as an eight-hour workshop, with four two-hour sessions, to fit into the time available in the participant's curricula or clinical practice. Each session was developed using three key principles: to provide evidence of the current deficiencies and the need for a new approach, to model the approach and allow participants to practice new skills. | **Screening:** not defined<br>**BI:** Brief Behavior Change Counseling (5As [Ask, Alert, Assess, Assist and arrange] plus MI) for tobacco and alcohol.<br>**Intervention described as brief:** yes.<br>**Intervention recipients:** patients attending primary health care clinics | **Mode of training:** Face to face by a nurse and psychologist trainer<br>**Duration:** 6 hours | **Outcomes:** Change in competency immediately after training and at 6 weeks. Competency was assessed using the Motivational Interviewing Treatment Integrity (MITI 3.1) tool, and a researcher designed fidelity tool.<br>**Findings:** Results showed a significant improvement in adoption of the guiding style (e.g. global score at baseline 2.0 (2.0–2.6) and in clinical practice 3.0 (2.7–3.3) p < 0.001) and completion of the 5A steps (e.g. assist score at baseline 1.26 (1.12–1.4) and in clinical practice 1.75 (1.61–1.89) p < 0.001). | NA |
| 4. | Malan 2015 [110] South Africa | Qualitative | **Setting:** Health facility<br>**Study populations:** health care workers.<br>**Sample size:** n = 41 (23 nurses, 18 doctors)<br>**Age and gender:** not provided | **BI:** Brief Behavior Change Counseling (5As [Ask, Alert, Assess, Assist and arrange] plus MI)<br>**Intervention described as brief:** yes.<br>**Intervention recipients:** patients attending primary health care clinics | **Mode of training:** Face to face by trained healthcare workers<br>**Duration:** 8 hours | None | **Outcome:** Healthcare workers' feedback on the training<br>**Findings:** Although the HCWs' confidence in their ability to offer the brief counselling improved, certain barriers to implementation were noted i.e. understaffing, lack of support from within the facility and poor continuity of care. |

*(Continued)*

**Table 4.** (Continued)

| | Author, year, country | Study design | Study population, Setting sample size; gender/age | Description of brief intervention | Training details | Quantitative outcomes and findings | Qualitative outcomes and findings |
|---|---|---|---|---|---|---|---|
| 5. | Rendall-Mkosi 2003 [41] South Africa | Qualitative | **Setting:** Health facility **Study populations:** healthcare workers. **Sample size:** n = 13 (9 nurses, 1 social worker, 1 doctor, 1 lay provider) **Age and gender:** not provided | Screening using CAGE and brief MI. **Intervention described as brief:** yes. **Intervention recipients:** patients attending primary health care clinics | **Mode of training:** Face to face by trained healthcare workers **Duration:** 7-hour training and 4-hour booster session | **None** | **Outcome:** Participants' perceptions on the training; Changes in participants' knowledge, skills, and attitude pre-and post-training **Findings:** Participants perceived the training as useful in building their capacity to offer alcohol brief interventions. |
| 6. | Morojele 2014 [42] South Africa | Qualitative | **Setting:** Alcohol serving establishments **Study population:** Lay counsellors. **Sample size:** n = 9 **Age: Range**- 28–41 years **Gender:** 6 female, 3 male | **Screening:** CAGE or TWEAK **BI:** 5 to 10 minutes MI focusing on alcohol **Intervention described as brief:** yes **Intervention recipients:** Bar patrons | **Mode of training:** Face to face by a registered MI trainer **Duration:** Two days plus one half-day booster | | **Outcomes:** Counselor perceptions on the training and intervention **Findings:** The counsellors reported that the training had been effective, informative, and had equipped them with skills to encourage behavior change. The counsellors all agreed that brief MI was ideal for use in the bars because it is of very short duration; and that the bar setting was appropriate for accessing people in need of interventions, Regarding their experiences in applying MI approaches, the counsellors reported having challenges in: emphasizing client responsibility (49%); asking for permission to screen and give feedback (34%). |

**Abbreviations:** AAR–Ask Advise Refer; 5As—Ask, Alert, Assess, Assist and Arrange; CAGE- Cut down, Annoyed, Guilty, Eye-opener; AUDIT- Alcohol Use Disorders Identification Test; BI- Brief Intervention; TWEAK–Tolerance, Worry about drinking, Eye-opener, Amnesia, Cut down on drinking.

## 3.5 Other BI outcomes

Six studies focused on various themes such as Knowledge of and practices related to BIs among health providers (n = 2) [113,114], cost-effectiveness analyses (n = 2) [115,116], and BI adaptation and development (n = 2) [117,118].

**3.5.1 Knowledge of and practices of BIs among healthcare workers.** Desalu et al. [113] explored the knowledge and practices of BI among physicians in Nigeria in a cross-sectional study and found that only 3.7% of physicians were using brief advice/counseling for smoking cessation. In a study conducted in Tanzania, investigating knowledge, attitudes, perceptions, and practices of ED healthcare personnel on alcohol use and a brief alcohol intervention Staton et al. [114] found that most participants felt comfortable asking patients (88.2%, n = 30) or

counseling patients (94.1, n = 32) about their risk with drinking; most participants agreed that it is common to ask patients about alcohol use behaviors (91.2%, n = 21), but only 41.2% (n = 12) reported always asking as part of their practice.

**3.5.2 BI adaptation and development.** Two studies conducted in South Africa reported on BI development and adaptation [117,118]. Carney et al [117] reported on the process of adapting a BI developed in the US (Teen Intervene) for the South African context. The adaptation process utilized the ADAPT-ITT framework [117]). During the adaptation process, the core components of the Teen Intervene intervention were retained, and adaptations focused on ensuring that the intervention fit the local context and was culturally appropriate [117].

Kekwaletswe et al. [118] in the early stages of developing a BI, sought perceptions of PLWH on the intervention. The authors sought patients' views on preferred facilitators; number and duration of sessions, session formats (group, individual sessions, videotapes, and written material); settings (i.e., clinic, school, community); involvement of family and friends, session content (tips on drinking less, coping without drinking etc.) The authors found that: The most preferred intervention facilitator was a fellow patient on ART; the most preferred number of sessions was two (40.3%); One hour was the most preferred session duration (40.8%); The most preferred format was the group session format (87.2% of participants). Most participants (95.1%) chose "clinic" as an appropriate setting for the proposed alcohol-focused intervention; Most participants (84.5%) favored having family members or friends in the sessions; "Sharing coping strategies that do not involve alcohol" was perceived as most useful content (84.8%) [118]).

**3.5.3 Cost-effectiveness of substance use BIs.** Two studies conducted a cost-effectiveness analysis of BI in Uganda [115] and South Africa [116]). Both economic evaluations were embedded within randomized controlled trials (RCTs). Judith et al. [115] conducted a costing study of the components of a RCT that compared a counselling intervention (two in-person sessions) plus booster sessions delivered twice via short text message (SMS) (technology arm); and counselling intervention (two in-person sessions) plus booster sessions delivered via live calls from counsellors (live call arm); and a control (leaflet with information on the effects of substance use). The authors found no significant intervention effects compared to the control. However, the cost of the booster sessions differed. Costs for the technology-delivered booster sessions were 2.5 to 3 times the cost per participant of the live-call delivered booster intervention for 1000 participants [115]. Dwommoh et al. [116] conducted a cost-effectiveness analysis of MI intervention compared to MI-PST and Control and found that MI or MI-PST delivered by lay counsellors were cost-effective strategies for the reducing SUD for emergency department patients compared to a control condition (leaflet providing information on the effects of substance use) [116].

## 3.6 Components and structure of BIs

**3.6.1 Single session.** All the single session interventions (n = 32) were based on MI techniques and education on substance use harms. Kane et al. [60] additionally included some coping skills training in their BI. For the studies that had indicated the duration of the BI, the time taken to deliver the session lasted from 5 minutes [70] to 3 hours [59]. See Table 1 for more details.

**3.6.2 Multisession BIs.** Of the multisession studies in which the number of BI sessions was provided (n = 28), most interventions were offered over two (n = 11), three (n = 8), and four (n = 6) sessions. Only three studies reported interventions longer than four sessions i.e., six (n = 2) [86,87] and seven sessions (n = 1) [92]). Of the studies in which the duration of each session was provided, the amount of time spent across all sessions ranged from 15 minutes ([96] to 4 hours ([76] (Table 2).

Of the studies reporting intervention content, most were based on MI only (n = 10), MI and problem-solving therapy (PST) (n = 8), and education on substance use harms and relapse prevention skills (n = 5). Other studies reported on interventions based on MI and education on harms (n = 4), advice to cut down and education on harms (n = 3), MI and relapse prevention skills (n = 1), and MI and Cognitive Behavioral therapy (CBT) (n = 1). One study reported on a seven-session intervention that incorporated multiple psychotherapy techniques including motivational enhancement therapy, cognitive therapy, behavior therapy, family therapy, relapse prevention, and exposure therapy [92]. See Table 2 for more details.

## 4. Discussion

The goal of this paper was to map the literature on substance use screening and brief interventions in Africa. This paper makes a major contribution to the BI literature in Africa. Prior scoping reviews on BIs in Africa have summarized findings on alcohol BIs and have focused on the sub-Saharan Africa region [119,120]. From our findings, most of the studies were conducted in Southern and Eastern Africa. This finding is similar to those reported in two scoping reviews conducted to map literature on BIs in Africa. Both studies found that most of the research had been conducted in Southern and Eastern Africa [119,120]. The paucity in research in other parts of Africa is concerning and we call on researchers in those regions to prioritize investigating BI in their regions particularly given the high prevalence of substance use problems throughout Africa [6].

We found a gentle rise in the number of BI studies in Africa over the years. The slow pace of substance use BI research could be attributed to general barriers to conducting research in Africa e.g. low experience with research methods, lack of funding and time, competing work priorities, and a lack of belief that research is valuable or will make a difference [121,122]. We call on governments and funders to prioritize substance use BI research given the high burden of substance use in Africa [6], and the potential of BIs to effectively address that burden [123].

Most of the BIs were alcohol focused with much less done on other substances such as tobacco, khat. We did not find any study that had tested a BI focused on other drugs such as opioids or cannabis. Most of the BIs had been conducted in healthcare settings among general adult patients and were delivered by healthcare workers. This is not surprising given that BIs were initially targeted for implementation in primary health care settings [124]. Several BIs were successfully delivered by lay-providers [45,53,59,71]. This strategy of task-shifting is an important one for addressing the shortage of mental health providers in Africa [125].

One important goal of this scoping review was to describe the components of BIs in Africa. We found that most of the interventions referred to as 'brief' were delivered over 1–4 sessions. This is consistent with the definition by Mattoo et al [36] that BIs last 1–4 sessions. Most of the interventions used MI and education on substance use harms. This is again consistent with the BI guidelines provided by authorities such as WHO [124]. Other BIs did not fit these well accepted definitions. For example, some BIs lasted six [86,87] and seven sessions [92], and utilized approaches such as PST [79,82,88], and CBT [98]. Such interventions could better be described as 'brief therapies' [37]. In order to standardize the substance use BI research, and make synthesis of findings more possible, we recommend that authors make a distinction between BIs and brief therapies in research.

Findings on the effect of BIs on substance use were mixed. This is consistent with several review papers that have found the BI effectiveness to be mixed. Tanner et al. [126] in a systematic review and meta-analysis of BIs delivered in general medical settings found that BIs had small beneficial effects on alcohol use. Sahker et al [127] in another meta-analysis found that BIs delivered in outpatient medical care had no effect on drug use (other than alcohol) [127].

The mixed findings could be because of patient level factors such as severity of substance use and comorbid mental disorder or implementation factors such as healthcare worker competency, suitability of implementation environment etc. Further the mixed results could be because of variation in research methods. Larger studies with rigorous designs such as RCTs are warranted to better understand the efficacy of BIs. Moreover, future RCTs should take a realist approach and focus on exploring the sub-populations and contexts in which BIs work.

Overall, the BIs were reported as feasible to implement from the perspective of policy makers, providers, and the intervention recipients. Barriers identified largely related to structural issues such as lack of time and space [47,49]; competing work priorities [48]; and lack of referral pathways [71]. These are factors that need to be considered during implementation of BIs to facilitate sustainability.

Despite the publishing of guidelines on delivery of BIs in primary health care more than a decade ago [124], our findings show that the practice of BIs within the primary health care setting in Africa remains limited. This is concerning because the primary healthcare setting represents an opportune place to have discussions about substance use. For example, during primary care consults, patients often present with both acute and chronic health conditions whose onset or progression may be exacerbated by alcohol use e.g. hypertension and diabetes. Additionally, in Africa, opportunities for treatment for SUDs are limited because of scarce health services and human resources [9]. Implementing BIs in primary health care could therefore be a practical and affordable way of increasing access to care. Our findings on limited implementation of SBIs in primary care mirror those documented in a WHO report. The report shows that in 2017, 70% of countries had no data on whether SBI for alcohol use was being conducted within the primary care setting. Of the countries with data, a majority of the countries (64%) reported that only 1–10% of primary healthcare facilities were routinely implementing SBI [128].

## 4.1 Research gaps

Studies investigating the cost-effectiveness of BIs compared to control conditions are limited in Africa. Data upon which policy makers can make decisions about resource allocation are therefore limited. Future work should focus on conducting economic evaluations for substance use BIs.

Several other gaps were identified in BI research in Africa that future work can focus on. From our scoping review, we found that few studies had investigated the effectiveness of training healthcare workers and lay-providers to deliver the BIs, effectiveness and feasibility of digitally delivered BIs, and effectiveness of BIs for drug use (other than alcohol). Despite a high burden of substance use among youth in Africa [129], little had been done to explore the effectiveness and feasibility of BIs for youth and adolescents. Other vulnerable populations that had been missed out on the BI work included incarcerated persons, and minorities such as the LGBTQ community. We did not find any study that had explored mechanisms of change for BIs among the African population. Lastly, it remains unclear which sub-populations the BI is most effective for. We call for researchers in Africa to address these gaps so that the continent can reap the full benefits of BIs.

## 4.2 Limitations

Because this was a scoping review, we did not undertake a meta-analysis nor detailed synthesis of the findings of studies included in this review. Additionally, we did not conduct a quality assessment of the studies. A second limitation is that in the absence of a consistent definition of BIs in the literature, we settled on that by Mattoo et al [36] which described BIs as those

delivered over 1–4 sessions. Our scoping review could therefore include interventions that other authors may describe as brief therapies [37]. Thirdly, we only included articles published in English or had an English translation available. We did so because the authors were all English speaking and were therefore unable to interpret the findings of papers published in another language. This eligibility criteria could have led to the exclusion of papers published from Portuguese speaking, Francophone, and Arab speaking countries in Africa. Nonetheless this review describes for the first time the breadth of existing literature on substance use BI in Africa, identifies research gaps, and provides important directions for future research.

## 5. Conclusion

The purpose of this scoping review was to map the research that has been undertaken on substance BIs in Africa. Most of the available studies had been conducted in Eastern and Southern Africa. BIs mostly ranged from 1–4 sessions and were based on MI and psychoeducational principles. The BIs had mostly been conducted in primary care settings by non-specialist health workers. Study findings suggest that BIs are feasible to implement across a broad range of settings. Findings on effectiveness vary. We identified research gaps including limited work done on BIs in North, Central, and Western Africa; paucity of BI research focusing on substances other than alcohol; and limited literature on BIs among vulnerable populations such as youth and prisoners. Future work should focus on these areas.

## Supporting information

**S1 Checklist. Completed PRISMA ScR 2009 checklist.**
(DOCX)

**S1 File. Search terms for PsychINFO database.**
(PDF)

**S2 File. Search terms for PubMed database.**
(CSV)

**S3 File. Search terms for CINAHL database.**
(PDF)

**S4 File. Search terms for Web of Science database.**
(PDF)

**S5 File. Search terms for PsychINFO database.**
(PDF)

## Author Contributions

**Conceptualization:** Florence Jaguga, Sarah Kanana Kiburi, Eunice Temet, Matthew C. Aalsma, Mary A. Ott, Rachel W. Maina, Juddy Wachira, Cyprian Mostert, Gilliane Kosgei, Angeline Tenge, Lukoye Atwoli.

**Data curation:** Florence Jaguga, Sarah Kanana Kiburi, Eunice Temet.

**Formal analysis:** Florence Jaguga, Sarah Kanana Kiburi, Eunice Temet, Gilliane Kosgei, Angeline Tenge.

**Investigation:** Florence Jaguga, Sarah Kanana Kiburi, Eunice Temet.

**Methodology:** Florence Jaguga, Sarah Kanana Kiburi, Eunice Temet, Matthew C. Aalsma, Mary A. Ott, Rachel W. Maina, Juddy Wachira, Cyprian Mostert, Gilliane Kosgei, Angeline Tenge, Lukoye Atwoli.

**Project administration:** Florence Jaguga, Sarah Kanana Kiburi.

**Supervision:** Florence Jaguga, Sarah Kanana Kiburi, Matthew C. Aalsma, Mary A. Ott, Lukoye Atwoli.

**Validation:** Florence Jaguga, Sarah Kanana Kiburi, Eunice Temet, Matthew C. Aalsma, Mary A. Ott, Rachel W. Maina, Juddy Wachira, Cyprian Mostert, Lukoye Atwoli.

**Visualization:** Florence Jaguga.

**Writing – original draft:** Florence Jaguga.

**Writing – review & editing:** Florence Jaguga, Sarah Kanana Kiburi, Eunice Temet, Matthew C. Aalsma, Mary A. Ott, Rachel W. Maina, Juddy Wachira, Cyprian Mostert, Gilliane Kosgei, Angeline Tenge, Lukoye Atwoli.

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
