## [Decision Letter · Decision Letter 0]

28 Jun 2024

PGPH-D-24-01083

A scoping review of substance use brief interventions in Africa

Dear Dr. JAGUGA,

Thank you for submitting your manuscript to PLOS Global Public Health. After careful consideration, we feel that it has merit but does not fully meet PLOS Global Public Health’s publication criteria as it currently stands. Therefore, we invite you to submit a revised version of the manuscript that addresses the points raised during the review process.

Thank you for submitting this manuscript reporting on a scoping review of substance use brief interventions in Africa. The review is well-conducted and adheres to the appropriate methodological guidelines and reporting standards.

A number of changes are suggested to improve the reporting of the methods used, and the presentation of the findings of the review. 

It would be important to specifically address the following comments:

- Please clarify the nuances between brief interventions and brief therapies. 

- Please address the points raised regarding the search strategy/methodology, and provide all search strategies as appendices. If there are limitations to the search strategy (i.e., no use of MEsH terms), please address these in the limitations section of the manuscript. 

- Please address the suggestions regarding the presentation of review findings and tables. 

We look forward to receiving your revised manuscript.

Kind regards,

Guillaume Fontaine, PhD, RN

Academic Editor

Journal Requirements:

Additional Editor Comments (if provided):

N/A

Reviewers' comments:

Reviewer's Responses to Questions

**Comments to the Author**

1. Does this manuscript meet PLOS Global Public Health’s publication criteria? Is the manuscript technically sound, and do the data support the conclusions? The manuscript must describe methodologically and ethically rigorous research with conclusions that are appropriately drawn based on the data presented.

Reviewer #1: Yes

Reviewer #2: Yes

2. Has the statistical analysis been performed appropriately and rigorously?

Reviewer #1: Yes

Reviewer #2: N/A

3. Have the authors made all data underlying the findings in their manuscript fully available (please refer to the Data Availability Statement at the start of the manuscript PDF file)?

Reviewer #1: Yes

Reviewer #2: Yes

4. Is the manuscript presented in an intelligible fashion and written in standard English?

Reviewer #1: Yes

Reviewer #2: Yes

5. Review Comments to the Author

Reviewer #1: This scoping review landscapes the literature on brief interventions for substance use disorders in Africa. This is a well written manuscript and I have some feedback that could help enhance it further.

A) Major comments

I have two fundamental concerns, one which can be corrected and one which cannot be corrected as it is beyond the control of the authors.

(1) 'For this study, BIs were defined as those delivered over 1-4 sessions whether the authors referred to them as brief or not'. Applying a numerical paradigm to define BIs based purely on number of sessions is problematic as the four sessions could be extensive and hence categorised under 'brief therapies' and not 'brief interventions'. I would suggest that the authors clarify in the paper that their inclusion criteria covers both brief interventions and brief therapies (provided they are less than 5 sessions). This is the piece that can be corrected as it is within the control of the authors.

2) Of the 974 abstracts that were screened almost 90% (n=872) were excluded. The reasons for exclusion include ineligibility or inability to retrieve the full text. It is unclear how many papers fall under the latter category, especially since the wording used indicates that most of those excluded were not retrievable ('We managed to retrieve 102 full texts'). It would be important to very clearly state how many were excluded because the papers were not retrievable as that has implications on the internal and external validity of your findings. This would not be correctable as this is beyond the control of the authors as they have already tried their best to retrieve the papers but were not successful

B) Minor comments

(1) Abstract: There is nothing in the results about the effectiveness or feasibility of the interventions.

(2) Please provide a rationale for conducting a scoping review and not a systematic review

(3) Please specify what study designs were eligible

(4) While scoping review protocols are not accepted on Prospero one can register them on other portals such as OSF

(5) In the tables that described the studies 'Study population, Setting sample size; gender/age' are all lumped together. They should be separated in discrete columns.

(6) Table 3 While the participants were the people who were trained, it is not clear what the target condition was i.e. who was the intervention delivered to. Please add a column to indicate that

Reviewer #2: General comments: Thank you for the opportunity to review the paper entitled: A scoping review of substance use brief interventions in Africa. This paper was interesting. Overall, I would recommend this manuscript for publication with major revision due to its significant flaws in the research method, especially regarding the literature search and data extraction. Please see below for the comments related to this paper.

Title: The title is adequate and reflect the content of the manuscript.

Abstract: The abstract is generally well-written. However, it would be necessary to rework it to include a problem statement/introduction, as it currently begins directly with the objectives of the scoping review. The abstract could also be revised to be more concise, especially in the methods section.

Introduction: The introduction does not define what substance use refers to. At line 41, "inject drugs" is mentioned, and at line 42, "alcohol-attributable disease" is referenced. Several terms are used in the search strategy (Supporting File 1) such as alcohol, tobacco, cigarette, cannabis. However, these elements should be better specified.

Line 54 should be more nuanced; while BIs can be used, this can be influenced by other factors such as the skills of healthcare professionals performing them, the contexts in which they are implemented, and other factors related to individuals using substances (e.g., social support, financial resources).

The objectives of the scoping review are relevant.

Methods: At lines 79-80, it is mentioned that "an exploratory search was carried out to determine the extent of literature on substance use BIs in Africa." However, no information is provided on how this exploratory search was conducted, the factors influencing the research question or objectives, or the selection of keywords used in the search strategy. Additionally, the databases used for this exploratory search are not specified.

At lines 87-90, the search strategy is briefly explained. It would be necessary to clarify why articles "published in English or translated to English" were included, considering that articles could be published in other languages. Supporting File 1 presents the search strategy in PsychINFO. However, the strategy does not adhere to expected database search practices. Specifically, no Medical Subject Headings (MeSH) appear to have been included, and keywords do not use truncation (i.e., *) or quotation marks to include exact terms. Additionally, several keywords or alternate terms are missing from each concept (e.g., e-cig, illicit substance, cannabinoids, genja, motivational intervention, risk reduction, harm reduction).

It is mentioned that there is considerable heterogeneity in the definitions of BI. However, the specific definition adopted by the authors is not explicitly stated.

The process for abstract screening is generally well-explained. However, it is not mentioned whether pilot testing was conducted to standardize this process and enhance agreement among reviewers. The full-text review process is well-described, but it should be clarified who the third author was that was invited to resolve conflicts.

Certain components of interventions are extracted (e.g., mode of delivery, number of sessions, duration). However, it would be necessary to extract all components of BIs to address the fifth objective of the scoping review, using the Template for Intervention Description and Replication (TIDieR).

Results: Overall, this section is well-executed, but modifications should be made to enhance readability and deepen the extracted data.

The trend of article publications is presented. However, this element seems to have limited relevance for addressing the objectives of the scoping review as indicated in lines 59-62.

A general description of the studies is provided at lines 155-158. However, it would be desirable to specify other more macroscopic results (e.g., design, targeted populations, type of BI, theories used) to offer a comprehensive overview of all 80 studies. It would be relevant to present sample sizes by type of study design (e.g., qualitative design X to X, cross-sectional study X to X, intervention studies X to X).

Tables 1, 2, and 3 contain a large amount of pertinent information. It would be necessary to add a list of abbreviations at the bottom of these tables.

At line 191, it is mentioned that thirty-two studies explored multi-session BIs. However, this number does not seem to match the subsequent text. Table 2 has a similar format to Table 1. However, the studies should be rearranged alphabetically, as in Table 1. Alternatively, both tables should be organized by study design (e.g., quantitative, qualitative, mixed) and then alphabetically.

Discussion: The discussion is interesting, but it would be helpful to provide a reminder of the research question and highlight the most interesting results before delving into the discussion of different themes. It would also be valuable to address elements related to other BI outcomes (e.g., cost-effectiveness, knowledge, attitudes, practices on BIs).

Research gaps: The section is interesting to provide avenues for reflection for researchers and stakeholders.

Limitations: It is mentioned "the aim of this systematic review" on line 331. However, this should be changed to "scoping review." Since the search strategy was not comprehensive, it is not appropriate to indicate that this scoping review used a systematic approach.

PRISMA Flowchart: An extra period is found after "Duplicates removed.". A number is missing after "Records after duplicates removed". There is a discrepancy between the exclusion reasons stated in the PRISMA and the manuscript ("Intervention not meeting our definition of brief", "Intervention not targeting substance use"). Additionally, the number of articles screened and included due to the hand search should be specified.

6. PLOS authors have the option to publish the peer review history of their article (what does this mean?). If published, this will include your full peer review and any attached files.

**Do you want your identity to be public for this peer review?** For information about this choice, including consent withdrawal, please see our Privacy Policy.

Reviewer #1: **Yes: **Abhijit Nadkarni

Reviewer #2: **Yes: **Billy Vinette

---

## [Editor Report · Decision Letter 1]

9 Sep 2024

A scoping review of substance use brief interventions in Africa

PGPH-D-24-01083R1

Dear Dr. JAGUGA,

We are pleased to inform you that your manuscript 'A scoping review of substance use brief interventions in Africa' has been provisionally accepted for publication in PLOS Global Public Health.

Best regards,

Guillaume Fontaine, PhD, RN

Academic Editor

Associate Editor Guillaume Fontaine: Thank you for addressing the reviewers' comments thoroughly and with great attention to detail. Your revisions have strengthened the manuscript, ensuring that it meets the high standards of PLOS Global Public Health. I look forward to seeing this work published.